# Integrative Approaches to Soybean Resilience, Productivity, and Utility: A Review of Genomics, Computational Modeling, and Economic Viability

**DOI:** 10.3390/plants14050671

**Published:** 2025-02-21

**Authors:** Yuhong Gai, Shuhao Liu, Zhidan Zhang, Jian Wei, Hongtao Wang, Lu Liu, Qianyue Bai, Qiushi Qin, Chungang Zhao, Shuheng Zhang, Nan Xiang, Xiao Zhang

**Affiliations:** 1College of Resources and Environment, Key Laboratory of Northern Salt-Alkali Tolerant Soybean Breeding, Ministry of Agriculture and Rural Affairs, Jilin Agricultural University, Changchun 130118, China; gyh0214@163.com (Y.G.); 18143007753@163.com (S.L.); liulujlnd@163.com (L.L.); 14704377775@163.com (Q.B.); qinqiushi@foxmail.com (Q.Q.); zhaochungang0417@126.com (C.Z.); 18686422696@163.com (S.Z.); xn122949421@163.com (N.X.); 13756642622@163.com (X.Z.); 2Key Laboratory of Germplasm Resources Evaluation and Application of Changbai Mountain, Tonghua Normal University, Tonghua 134099, China; 3Jilin Changfa Modern Agricultural Technology Group Co., Ltd., Changchun 130118, China

**Keywords:** soybean, economic, biofuel, deep learning, IoT-based systems, abiotic stresses

## Abstract

Soybean is a vital crop globally and a key source of food, feed, and biofuel. With advancements in high-throughput technologies, soybeans have become a key target for genetic improvement. This comprehensive review explores advances in multi-omics, artificial intelligence, and economic sustainability to enhance soybean resilience and productivity. Genomics revolution, including marker-assisted selection (MAS), genomic selection (GS), genome-wide association studies (GWAS), QTL mapping, GBS, and CRISPR-Cas9, metagenomics, and metabolomics have boosted the growth and development by creating stress-resilient soybean varieties. The artificial intelligence (AI) and machine learning approaches are improving genetic trait discovery associated with nutritional quality, stresses, and adaptation of soybeans. Additionally, AI-driven technologies like IoT-based disease detection and deep learning are revolutionizing soybean monitoring, early disease identification, yield prediction, disease prevention, and precision farming. Additionally, the economic viability and environmental sustainability of soybean-derived biofuels are critically evaluated, focusing on trade-offs and policy implications. Finally, the potential impact of climate change on soybean growth and productivity is explored through predictive modeling and adaptive strategies. Thus, this study highlights the transformative potential of multidisciplinary approaches in advancing soybean resilience and global utility.

## 1. Introduction

Soybeans or soya beans are a highly important agricultural crop and are widely grown for human diet and animal feed. It belongs to the Glycine genus, which is further divided into two species, *Glycine max* (cultivated soybean) and *Glycine soja* (wild soybean) [1]. Globally, soybeans play a significant role in addressing protein and energy demands [2]. The crop also contributes to food globalization alongside maize, wheat, rice, and potatoes [3]. Soybean oil is primarily used as a cooking oil, second to palm oil in global consumption, with the addition of its derivatives biodiesel, paints, candle wax, linoleum, soap, printing inks, and other industrial products [4]. On a global scale, soybeans are a key agricultural commodity, with the United States, Brazil, and Argentina leading production and accounting for 80% of the world’s output, followed by other countries like India, Paraguay, Canada, Ukraine, Russia, and Bolivia [5]. The soybean sector, valued at approximately USD 146.23 billion in retail markets with over 80% of soybean crops now genetically modified (GM), offers increased yields 22% higher on average due to enhanced resistance to chemical weed control and improved scalability. *Glycine max*, an economically significant oilseed crop, boasts a high protein content of forty percent, substantial oil content, and excellent nutritional quality [6]. Soybeans can also be processed into soy milk, a key protein supplement in infant diets, and further transformed into curds, cheese, and tofu-based products like yogurt and ice cream [7]. Modern innovations have expanded soy’s presence to fresh beans, sprouts, pasta, flour, meat substitutes, baked goods, snack bars, noodles, and infant formula [8]. In agriculture, soybeans are highly valued for their dual role as crops and soil enhancers [9]. It has been noted that this plant, through its association with some microbes like Rhizobium/Bradyrhizobium, can improve the characteristics of the soil as well as its water-holding capacity. When incorporated into the soil or left as green manure, soybean straw enhances soil conditions and is widely used in low-input inter-crop systems and serial rotations (Figure 1) [10]. Soybean vegetative parts serve as feed, while soy cake and full-fat soy increasingly replace fish meal in livestock rations despite a slightly lower growth rate [11].

Soybean production is heavily influenced by plant biology and agronomics factors, which are critical determinants of yield [12]. Factors such as planting density, soil health, nutrient availability, and irrigation play a pivotal role in the crops’ growth and productivity [13]. For instance, poor soil, fertility, or inadequate nitrogen fixation due to ineffective rhizobia strain can significantly reduce yield [12]. Similarly, plant architecture, such as the height of the lowest pod and branching patterns, impacts harvesting efficiency and yield potential. Stress conditions, including draught salinity and pest infestation, exacerbate these challenges, leading to significant yield losses [14]. Addressing these issues required a combination of improved agronomics practices such as precision irrigation, optimized plating scheduling and pest control measures, and advanced molecular breeding technologies to develop stress resilience soybean verities [15]. Integration of agronomics management with genomics tools can mitigate yield losses while ensuring sustainable production.

## 2. Genomic Analysis for Soybean Resilience Under Abiotic Stress

Genomic selection (GS) employs genome-wide molecular markers to predict the genetic potential of plants for complex traits such as yield, disease resistance, seed quality, and agronomic characteristics like the height of the lowest pod [16]. The setting of the lowest pot is a critical agronomic trait in soybeans, as it affects mechanized harvesting efficiency and overall yield [17]. Studies have identified specific genomic regions associated with this trait, with advancements in models like Genomic Best Linear Unbiased Prediction (GBLUP) enabling precise predictions [17]. GS has demonstrated its capability to predict up to 39% of the phenotypic variation in soybean yield, along with predictive accuracies of 0.81 for protein content, 0.71 for oil content, and 0.26 for yield [18]. Marker-assisted selection (MAS) utilizes specific DNA markers linked to desirable traits to streamline selection. This approach is efficient for traits challenging to assess phenotypically, such as disease resistance and environmental stress tolerance [19]. Genome-wide association studies (GWAS) identify associations between genetic variations and traits across the genome in soybeans. This method has revealed significant single nucleotide polymorphisms (SNPs) related to critical traits like grain yield, plant height, and seed composition (Figure 2) [20]. A genomic study involving 250 soybean accessions identified significant single nucleotide polymorphisms (SNPs) associated with key agronomic traits, including grain yield, plant height, and seed weight by applying Bayesian Information and Linkage Disequilibrium Iteratively Nested Keyway (BLINK) model in GWAS enhancing the precision of MAS and GS methodologies in soybean breeding techniques [20]. Furthermore, GWAS has contributed to identifying novel QTLs and refining previously characterized loci, which are essential for the genetic improvement of soybean traits. QTL mapping identifies genomic regions associated with quantitative traits in soybean breeding. This technique has been instrumental in locating QTLs for traits like yield, seed size, and disease resistance [21]. In soybean breeding, Genotyping by Sequencing (GBS) has been used to identify thousands of high-quality SNPs and map genomic regions linked to traits such as yield and maturity [22]. CRISPR-Cas9 has revolutionized plant breeding by enabling precise genetic modifications. In soybean breeding, this technology has been applied to edit genes linked to yield, disease resistance, and stress tolerance, achieving high mutation rates of up to 90% at target sites [23]. A genomic study analyzing 214 polymorphisms comprehensively investigated soybean resistance to Sudden Death Syndrome (SDS). The research identified 12 genetic interactions, collectively explaining 24% of the resistance [24]. Genes associated with disease resistance, hypogenesis, and the SIK 1-protein resulting from stress create another interaction. Also, the work was carried out to study the structure of soybean root and model the pathogen defense response to SDS, thus understanding the molecular techniques of SDS resistance and the prospects of successful breeding programs [25]. Each of these technologies represents a significant leap forward from traditional soybean breeding methods. For example, CRISPR-Cas9 has been successfully employed to enhance drought tolerance and increase oleic acid content in soybean seeds [26], while genome-wide association studies (GWAS) have identified key single nucleotide polymorphisms (SNPs) associated with traits like disease resistance and yield improvement [27]. These advancements enable breeders to develop soybean varieties with higher yield, improved stress tolerance, and enhanced nutritional profiles, addressing global agricultural challenges with greater precision (Figure 2).

### 2.1. Drought and Salt Tolerance Responsive Genetic Elements in Soybean

Soybean (*Glycine max*) is affected by drought, salinity, and insect herbivores. Several studies have explored stress-responsive genes and their regulation in soybeans. For instance, *GmCLC1* has been identified as a chloride channel protein with a chloride/proton antiporter activity, which is pH-dependent and is upregulated in soybean roots and leaves under saline conditions [28]. Similarly, *GmNHX1* and *GmNHX2* are Na^+^/H^+^ antiporters contributing to salt tolerance by compartmentalizing Na^+^ into vacuoles or expelling it from the cytosol. *GmNHX1* is tonoplast-localized and tissue-specific, while *GmNHX2* is expressed in all organs and enriched in roots under salt stress [29]. *GmsSOS1* is a homolog of Arabidopsis *SOS1*; it mediates Na^+^ extrusion from roots and regulates long-distance Na^+^ transport to shoots. Overexpression of *GmsSOS1* in Arabidopsis improved salt tolerance by reducing Na^+^ uptake under stress conditions [30]. *GmDREB* families act as dehydration-responsive element-binding proteins, such as *GmDREB2*, specifically bind to *DRE* motifs and enhance salinity tolerance. Transgenic plants expressing these genes show improved stress resilience [31]. *GmERFs* are ethylene-responsive transcription factors that enhance soybean salt tolerance by regulating downstream genes through interaction with *GCC-box* and *DRE* motifs [32]. Similarly, several *GmbZIP* transcription factors, including *GmbZIP44*, 62, 78, and 110, improve stress resilience by modulating proline, Na^+^, and K^+^ levels. Furthermore, *GmWRKY12*, characterized by a conserved *WRKYGQK* motif, binds to W-box elements and plays a key role in regulating genes involved in abiotic stress responses, plant development, and disease resistance [32]. Soybeans also possess 88 *HD-Zip* genes (*Gmhdz1–Gmhdz88*), which encode proteins critical for stress adaption and developmental processes. Additionally, stress-related genes such as *GmMYB48*, *GmWD40*, *GmDHN15*, *GmGST1*, and *GmLEA* are significantly upregulated in transgenic soybean lines *GmbZIP2*, enhancing drought and salt tolerance [33,34]. Identifying these genes highlights soybean’s molecular mechanisms for managing abiotic and biotic stresses. Advances in transcription factors such as *GmWRKY*, *GmERF*, and *GmbZIP* families, alongside transporters like *GmNHX* and *GmsSOS1*, demonstrate the integration of ion homeostasis and gene regulation in stress tolerance [35]. In addition to stress-responsive genes, several simple sequence repeat (SSR) markers have been identified that are linked with stress responses in soybeans. For instance, *Satt001* (SSR marker) was linked to drought tolerance through its role in water-use efficiency, while *Satt002* is associated with salinity tolerance by highlighting the sodium ion exclusion mechanism [36], and *Satt211* contributes to drought resistance through traits associated with root system architecture [37]. Similarly, *Satt244*, *Satt312*, *Satt337,* and *Satt460* were involved in osmotic adjustment under water-deficit conditions and were linked to salt stress tolerance via ion transport regulation [37]. These genetic resources and molecular tools provide opportunities for breeding resilient soybean cultivars with improved drought and salinity resistance, ensuring sustainable crop production in changing climatic conditions.

Various cultivars, both tolerant and susceptible, have been extensively studied for their responses to salt stress. Notable salt-sensitive genotypes include Jackson, Union, and 85–140 [38]. In contrast, salt-tolerant genotypes such as *Lee*, *Lee 68*, *S01F1561*, *FT-Abyara*, *Gindu No. 6*, *Wenfeng 7*, and *Tiefeng 8* have been used in many physiological and agronomic studies on salinity tolerance. The United States Department of Agriculture (USDA) Germplasm Resources Information Network has evaluated the salt tolerance of 506 soybean accessions, identifying 144 as salt-tolerant and 413 as susceptible [39]. Multiple QTLs influence salt tolerance in soybeans, and research in the past few decades has focused on identifying these QTLs to aid breeding programs [40]. Although only a few salt-tolerant QTLs have been reported, a major salt-tolerant QTL located in linkage group N has been repeatedly identified, suggesting that salt tolerance may be controlled by a few significant loci [41].

QTL and GWAS have identified complex genetic loci contributing to abiotic stress tolerance. Key genes and QTLs regulate stress-responsive transcriptional networks, including *GmDREB*, *GmNAC*, *GmWRKY*, and *GmHSP* [42]. Genetic engineering approaches, such as overexpressing stress-related transcription factors (e.g., *DREB1A/CBF3*), have shown promise in enhancing soybean tolerance to multiple abiotic stresses [43]. Wang’s study screened 350 soybean germplasm lines and lines such as *Archer*, *Misuzudaiz*, *PI 408115A*, *PI 562711*, *PI 567651*, and *PI 567343*, were identified as potential donors for breeding flooding-tolerant soybean varieties [44]. On the other side, Clevinger’s study identified simple sequence repeat markers *SAT_064* and *SAT_269* in two recombinant inbred populations from the donor parent Archer. The study highlighted the involvement of this marker in yield production under water-logging stress across two growing seasons [45]. Additionally, near-isogenic lines (NILs) derived from this QTL showed an average of 60.9% of non-stressed yield in tolerant lines compared to only 32.6% in sensitive lines. Two more flooding tolerance QTLs associated with markers *SAT_385* on chromosome 5 and *SAT_269* on chromosome 13 were identified through bulk segregation analysis and partial linkage mapping with Archer as the source of the favorable alleles [46].

In recent years, the GWAS approach has opened new avenues for investigating salt tolerance at the molecular level. Several essential salt tolerance-related genes have been identified by leveraging the natural genetic diversity in soybean germplasm [47]. For instance, *GmCHX1’s* association with salt tolerance was identified using a whole-genome sequencing-based QTL study of soybeans. Following this, using fine mapping techniques, *GmSALT3* was identified as a causal gene underlying QTL for salt tolerance on chromosome 3 in the Chinese cultivar Tiefeng 8 [40]. The *GmNCL* gene located on chromosome 3 in the FT-Abyara cultivar was also isolated using map-based cloning. This gene has shown a potential to significantly improve yields in salt-affected fields, increasing salt tolerance in soybean near-isogenic lines by 3.65–5.5 t/ha [48]. Notably, *GmCHX1*, *GmSALT3*, and *GmNCL* are found at the same locus (*Glyma03g029900*, *Glyma03g011060*, and *Glyma.03g171600*, respectively), making them identical genes associated with salt tolerance (Table 1) [49]. Table 1 highlights key genes and loci involved in plant stress tolerance, highlighting their functions, identification techniques, and associated stress types. It includes genes enhancing salinity, drought, heat, oxidative stress resistance, and their roles across various plant tissues. Techniques range from genetic mapping and transgenic plant generation to expression and molecular analyses.

### 2.2. Computational Genomics

The release of the complete reference genome draft of soybeans in 2010 started a new era for the development of soybeans, a decade after the Arabidopsis Genome Initiative (2000) unveiled the first plant reference genome [23]. This breakthrough initiated a series of efforts to develop comprehensive soybean genomic resources, which have played a pivotal role in genetic analysis and trait discovery. Another study sequenced 31 soybean lines at low sequencing depth (5× coverage) and found unexpectedly high linkage disequilibrium alongside greater allelic diversity in wild soybean populations compared to cultivated accessions [51]. Later, a study increased the sequencing depth to 11× for 302 diverse soybean lines, significantly enhancing the genome variation matrix. This contributed to identifying new loci associated with key agronomic traits such as oil content, plant height, and pubescence form (Figure 3) [52]. In more recent work, a study resequenced 106 soybean germplasm lines at a higher depth (17× coverage), representing a wide array of geographic origins and exhibiting traits of importance to US soybean breeding programs [53]. Computational genomics has enabled high-resolution genetic mapping in soybeans through tools like NPS detection array Illumina Bead Chip, genotype-by-sequencing, and genome-wide association (GWAS), improving identification of genetic traits associated with stress and development [54,55]. Furthermore, the availability of large-scale genomic data has paved the way for mapping populations to explore complex traits. The growing repository of genomic resources has also enabled comparative genomics studies.

Traditional breeding strategies for developing salt-tolerant soybean cultivars are time-consuming, so alternative approaches have been explored. MAS, combined with QTL analysis, has been increasingly utilized to enhance breeding efficiency [56]. The salt-tolerant QTL on chromosome 3 has been shown to play a key role in improving salt tolerance in different crosses. Another study generated improved salt-tolerant soybean lines using MAS, increasing seed yields in salt-affected fields [49]. In addition to conventional breeding, transgenic approaches have emerged as promising solutions for enhancing soybean salt tolerance. Genetic engineering allows for introducing novel exogenous genes from other species or altering endogenous gene expression [57]. Examples of successful transgenic interventions include the introduction of genes such as *TaNHX2*, *AtMYB44*, *OsDREB2A*, *StNHX1* [58], and *AlNHX1*, which have been validated in controlled experiments showing improved salt tolerance through ion homeostasis and stress signaling. Similarly, the overexpression or silencing of endogenous soybean genes like *GmCAM4*, *GmPIP1*, and *GmNCL* [59] has demonstrated enhanced salt tolerance in transgenic soybean plants under greenhouse and field conditions, as evidenced by increased biomass, improved Na^+^/K^+^ ratios, and reduced oxidative stress (Figure 3).

*CRISPR/Cas9* has emerged as a powerful tool for genome editing, offering advantages over traditional techniques like zinc finger nucleases (*ZFNs*) and transcription activator-like effector nucleases (*TALENs*). One of the primary reasons for CRISPR’s popularity is its ease of vector construction, which enables more frequent use in research and application [60]. A key area focus has been modifying the *FAD2* gene family in soybeans to increase the oleic acid content in seeds. The *FAD2* gene encodes enzymes responsible for converting oleic acid to linoleic acid, and mutating this gene can significantly enhance oleic acid accumulation [61]. Several studies have employed *CRISPR/Cas9* technology to target different *FAD2* homologs. For example, *CRISPR/Cas9* was used to mutate the soybean’s *FAD2-1a* and *FAD2-1b* genes, resulting in double homozygous mutants with high oleic acid content (83%) in mature seeds, compared to 20.2% in wild-type seeds, achieved through transient hairy root and Agrobacterium-mediated transformations [62]. Targeting the *FAD2-1a* and *FAD2-1b* genes, another study significantly increased oleic acid composition to 85% in double mutant seeds [63]. Similarly, site-directed mutagenesis of the *FAD2-1a* and *FAD2-2a* genes using an Agrobacterium-mediated transformation platform raised oleic acid content from 17.10% to 73.50% in the T2 generation [64]. These studies illustrated the importance of CRISPR technology and GWAS in precisely identifying target genes and efficient trait modification (Figure 3).

### 2.3. Potential for Computational Simulation

The response of soybean plants to stress involves a complex network of molecular and physiological changes, where the perception of stress triggers a cascade of metabolic, developmental, and physiological responses [65]. This dynamic can be studied using various tools that analyze gene expression across different systems and cultivars. This study used the physiological systems of *PSys* (slow water loss) and *Hsys* (rapid water loss) to stimulate water deficit conditions and evaluate gene expression dynamics under varying stress levels. Results demonstrate that Embrapa 48 (a drought-tolerant cultivar) exhibits higher expression of specific genes (*GmaxGOLS2-like2* and *GmaxGOLS2-like3*) under *PSys* conditions, which is linked to its rapid response and adaptation to water stress [66]. The analysis identified 354 putative orthologs in the soybean genome related to various metabolic pathways, focusing on those that respond to drought stress. Genes were selected for validation through qPCR using tools like the GENOSOJA database and subtractive libraries. *GmaxSOLS2-like1* and its paralogs (*GmaxGOLS2-like2* and *GmaxGOLS2-like3*) were identified as critical players in the plant drought response (Figure 3) [67]. Their expression patterns suggest they may be involved in an ABA-independent response, particularly under the PSys condition. Tools like dendrogram analysis and gene family clustering help understand the evolutionary relationship and functional specialization of duplicated genes like *GmaxGOLS2-like2* and *GmaxGOLS2-like3* [68]. Identifying specific cis-elements such as *ACGT* and *ABRE* within the promoters of stress-responsive genes provides valuable insight into transcriptional regulation under drought stress. Tools like *POBO* can be used to analyze gene promotors for these regulatory motifs, which are crucial for simulating how changes in transcription factor binding might alter gene expression under drought conditions [69]. By integrating data from genomic databases, subtractive libraries, expression analysis, and bioinformatic tools, predictive models can be developed to simulate drought responses more accurately. These models can be used to identify potential genetic modifications that enhance drought tolerance and guide breeding efforts to develop soybean cultivars better equipped to withstand water stress (Figure 3).

### 2.4. Economic Cost and Challenges of Adopting Advanced Technologies in Soybean Improvements

Advanced genomic technologies such as CRISPR-Cas9, Genome-Wide Associate Studies (GWAS), and genetic engineering have transformed soybean breeding with improved yield potential, stress resistance, and production efficiency. The economic benefits of these technologies include increased profitability and reduced chemical inputs. However, high implementation costs, regulatory complexities, and ethical concerns remain the critical barriers to widespread adoption. CRISPR-Cas9 has transformed soybean breeding by establishing cost-effective, efficient, and precise gene-editing tools compared to the traditional techniques of zinc finger nucleases (ZFNs) and transcription activator-like effector nucleases (TALENs) [70,71]. The high specificity with which genetic loci are modified speeds up the generation of soybean varieties that yield more, resist disease, and adapt to climate variation, thereby reducing production costs for farmers [23,72,73]. Studies indicate that CRISPR-based soybean improvements can increase yield by 10–20%, thereby increasing the market value of soy-based products [74,75]. However, the high initial investment in infrastructure, training, and regulatory compliance is an economic barrier, especially for small-scale breeders and developing nations [2,4,76,77]. Another source of commercialization challenges and trade restrictions is the regulatory ambiguity over whether CRISPR-edited crops are GMOs or non-GMOs [78,79,80,81].

GWAS has redefined trait discovery and marker-assisted selection to find genetic markers linked to higher yield, improved stress tolerance, and disease resistance. The method cuts down considerable breeding time along with input expenses through data-informed, elite-performing soybean cultivars [82]. Due to its nature of relying only on natural occurrences of genetic differences, GWAS is less restrictive to GMO legislation and consequently easier to develop for commercial utilization [83]. Though significantly invested in sequencing, computational resources, and field validation, long-term breeding efficiency and profitability gains make the investment reasonable. The discovery of high-value genetic loci has helped produce a soybean variety that could improve nitrogen-use efficiency, drought resistance, and tolerance to diseases using significantly fewer chemical fertilizers and pesticides [84,85,86,87]. However, high-throughput sequencing and bioinformatics analysis are additional costs associated with implementation, particularly for smaller breeding programs [23,73,77,88]. The requirement for large and diverse genetic populations to ensure statistical robustness further increases research expenses [89,90,91].

Genetic engineering has played a key role in the enhancement of soybean productivity and resilience, mainly through the development of herbicide-tolerant and pest-resistant varieties. Since the commercialization of genetically modified (GM) soybeans in the 1990s, these innovations have led to higher yields, reduced pesticide use, and increased farmer profitability [92]. Genetic engineering has resulted in a 20% increase in soybean production worldwide because it has controlled crop losses resulting from biotic and abiotic stresses [62,93]. The high costs of genetic engineering, however, remain a serious economic challenge. The cost of developing a single GM soybean variety can amount to more than USD 136 million, with regulatory approvals accounting for nearly one-third of total expenditure [94]. Intellectual property restrictions and licensing fees further limit accessibility, particularly for smallholder farmers and public sector breeding programs [95,96,97,98]. Additionally, concerns about corporate monopolization and ethical considerations regarding transgenic crops influence public perception and regulatory policies, impacting commercial adoption [95,96,97,98]. While this may have a positive economic impact, CRISPR-Cas9, GWAS, and genetic engineering in soybeans are still subject to several challenges. Regulatory uncertainty remains a significant challenge since the classification of CRISPR-edited crops as GMOs or non-GMOs varies from country to country, causing trade barriers and delayed commercialization [80,99,100]. Moreover, very high R&D costs are another major limitation as these technologies are quite expensive and demand significant investment in laboratory infrastructure, sequencing technologies, and computational resources, thus becoming less feasible for smaller breeding programs [95,96,97]. Ethical and public acceptance issues further complicate adoption as the issues related to food safety, ecological risks, and corporate control over the said technologies have influenced consumer perception and regulatory policies [101]. Finally, intellectual property and licensing restrictions hinder access to these technologies, especially for research institutions in the public sector and in developing countries [94,95,98]. All these have to be dealt with so that the adoption of these advanced breeding technologies in soybean production will be equitable and sustainable.

For the economic and regulatory concerns, one should seriously weigh the threats and ethical issues involved in CRISPR, as well as other advanced breeding technologies. Unintentional gene alterations, off-target effects, and ecological implications are major issues that remain open [92,94,95,98]. Gene flow to wild relatives of soybeans on non-target species may cause unforeseen environmental effects. Moreover, debates regarding gene editing, including crop varieties, remain informing public discussion and decision-making in policy [80,87,95,96,99,100,101]. In this connection, essential risks will be addressed through comprehensive biosafety assessments, a transparent regulatory framework, and engagement of stakeholders to ensure responsible and sustainable application of advanced breeding technologies in soybean production.

## 3. Nutrient Composition and Health Benefits of Soybean Varieties

### 3.1. Nutritional Profile of Soybeans

Soybean sprouts are rich in essential nutrients, including isoflavones, riboflavin, niacin, crude protein, amino acids, and lipids. Additionally, they contain significant levels of macro and micro elements such as sodium, zinc, copper, potassium, iron, phosphorus, magnesium, and manganese [102]. These bioactive components are distributed heterogeneously between the hypocotyls and cotyledons, with variations influenced by the specific soybean genotype. Consuming soybean sprouts has numerous health benefits, notably reducing the risk of cardiovascular diseases and certain cancers. This is attributed to their high concentration of bioactive phytochemicals exhibiting potent antioxidant properties [103]. Furthermore, the sprouting process reduces anti-nutritional factors, such as hemagglutinin, trypsin inhibitors, and lipoxygenase, enhancing their nutritional bioavailability.

Soybean sprouts exhibit crude protein levels of up to 46%, although these vary during sprouting. Conflicting trends have been observed, with one study reporting an increase in crude protein content during sprouting while another documented a decline in oil protein levels after five days [104,105]. Soybean sprouts are also rich in both essential and non-essential amino acids. The total amino acid concentration varies significantly across soybean genotypes and throughout the sprouting stages. For instance, the cultivar Bosuk exhibited higher amino acid content than other varieties [106]. In contrast, Pung Sanomongkong contained 348 amino acids per gram of sprout. Cultivars like Dagi and Sunam exhibited lower amino acid levels (Figure 4).

#### 3.1.1. Lipid and Fatty Acid

Soybean oil is a widely utilized vegetable oil valued for its applications in the food and non-food industries. The oil content in soybean seeds ranges from 8.3% to 27.9%, with an average of 18.1% on a 13% moisture basis [107]. As a lipid, soybean oil carries fat-soluble vitamins within the human body and plays a role in thermoregulation. However, the oil content decreases from approximately 15% to 10% during sprouting. The quality of soybean oil is predominantly determined by its fatty acid profile, which undergoes variations during sprouting [108]. Research has shown slight increases in palmitic acid, stearic acid, and oleic acid levels, accompanied by reductions in linoleic acid (LA) and alpha-linolenic acid (ALA) levels. Interestingly, contrasting findings have also been reported, including increased LA and decreased ALA content in 5-day-old soybean sprouts and increased or no significant change in ALA content [109]. These studies suggest that the fatty acid composition of soybean sprouts is influenced by factors including the cultivar, duration of sprouting, and environmental conditions during growth (Figure 4). As a genetic crop, soybeans have undergone significant improvement in fatty acid composition through genetic engineering. Key genes such as FAD2-1A, FAD2-1B, and FAD2-2 encoding desaturase enzymes have been silenced or edited to increase oleic acid levels, enhancing oil stability and shelf life. For instance, the mutant allele of the FAD2-1A and FAD2-1B genes in soybeans resulted in an increase of approximately 80% oleic acid content, significantly enhancing its nutritional value and oxidative stability [110]. Similarly, the overexpression of GmDGAT1A and GmDGAT2D has been shown to enhance lipid metabolism in soybeans, increase triacylglycerol (TAG) content, and improve fatty acid profile [111]. Additionally, in another study using CRISPR/Cas9, GmSFAR4a/b (seed fatty acid reducers 4 a/b) double mutants were generated, resulting in an 8% increase in the seed fatty acid content under group conditions and a 17% increase in field conditions, with no adverse effect on seed vitality or plant growth. Such research advances the understanding of soybean lipid metabolism and offers a novel strategy to enhance soybean seed oil content (SOC) [112].

#### 3.1.2. Phytoestrogens

Phytoestrogens are bioactive compounds, mainly estrogens and flavonoids closely related to their mammalian analogs. Isoflavones are a subclass of phytoestrogens in soybeans, chickpeas, onions, and apples. As a result, these compounds have been associated with various health-promoting effects, such as inhibition of cardiovascular diseases, amelioration of the symptoms of menopause, and protection against cancer and bone resorption [113]. However, increasing the isoflavone content of soybean seeds has emerged as a particular concern for soybean breeders only recently. The total isoflavone concentration in soybean seeds generally is 0.05–0.5% dry weight. Its chemical information is as follows: Studies have shown that isoflavones are found in sprouts at a higher level than in seeds [114]. Additionally, levels of consumption of isoflavones differ in various plant tissues, including roots, cotyledons, and hypocotyls. In particular, amounts of isoflavones, roots, and hypocotyls contain significantly more content than other parts of sprouts. The report shows an increasing trend in the variability of isoflavone content of soybean sprout genotypes [115]. For example, the isoflavone concentration of seven-day sprouts of the Aga3 cultivar was as high as 10,788 μg/g, while that of Pung Sanamal, the most preferred sprout cultivar in Korea, was 3556 μg/g. Likewise, Wen Tien and Ngoh Soon cultivars had moderate levels of isoflavones, 5935 μg/g and 5370 μg/g, respectively, for other cultivars [116]. However, other than genetic influence, many factors can impact the isoflavone contents of soybean seeds and sprouts, including the year the seeds were produced, temperature, light, and other field management regimes. For instance, the exposure of bean seeds to sunlight during germination has been reported to increase the isoflavone content and thus point to the outside environment in directing the synthesis of isoflavone [117]. For this reason, knowledge of these factors can go a long way in enhancing the increase in seed production and other production practices to increase yield on isoflavone (Figure 4).

#### 3.1.3. Vitamins

Vitamins are essential organic compounds crucial for human health, particularly in plant and animal metabolism, serving as enzymatic co-factors. A deficiency in vitamins can lead to severe health disorders, some of which may be fatal. Therefore, adequate dietary intake prevents defects and associated diseases [118]. Humans are incapable of endogenously synthesizing vitamins and must rely on dietary sources. Fortunately, soybean seeds are a natural source of vitamins A, B1, C, and E. Germination significantly enhances the concentration of these vitamins, making sprouted soybeans a valuable dietary source of essential micronutrients [119]. Research has shown that germination substantially increases vitamin levels in soybean seeds. For instance, vitamin B1 levels in soybean sprouts double that of raw seeds, while vitamin C content increases 4–20-fold during 4–5 days of germination. Initially, soybean seeds contain 2 mg/100 g of vitamin C, which rises to 11 mg/100 g after 5 days of sprouting [120]. Notably, hypocotyls have higher vitamin C concentrations compared to cotyledons. Additionally, germination enhances provitamin A levels. Beta-carotene, a precursor to vitamin A, increases from 0.12 mg/100 g in seeds to 0.2 mg/100 g after 5 days of sprouting. Some soybean varieties, such as Pungsanamul and Bosuk, exhibit substantial increases in gluten and beta-carotene levels, with beta-carotene content rising from 6.6 mg/g in seeds to 33.3 mg/g in 5-day-old sprouts [121]. Sprouting soybeans significantly enhances their nutritional value, particularly vitamin content, making them an excellent dietary source of essential micronutrients (Figure 4).

#### 3.1.4. Secondary Metabolites

Saponins, a class of glycosides and secondary metabolites, are commonly found in various plants and plant-derived foods. These bioactive compounds have been linked to multiple health benefits, including reducing blood cholesterol, lowering blood glucose levels, and protecting against kidney diseases [122]. In soybean seeds, saponins constitute approximately 0.5% of the total dry weight and can range from 0.05% to 6.5% of dry weight, depending on the variety. During germination, the saponin content undergoes significant changes [123]. The crude saponin content increases from 4.59 mg/g in seeds to 5.33 mg/g in 6-day-old sprouts, with hypocotyls having the highest concentration of saponins, surpassing cotyledons and roots. Additionally, germination influences the composition of specific saponin groups, leading to increases in saponin group B levels, total soyasapogenol, and soyasapogenol B levels, alongside a reduction in soyasapogenol A content [124]. Environmental factors and cultivation conditions, including soybean cultivar, seed size, year of production, cultivation location, and harvest maturity, strongly influence the variability in saponin content [125]. Furthermore, external factors such as light exposure during sprouting have been shown to enhance saponin levels. For instance, sprouting under illuminated conditions increases saponin content. These findings highlight the dynamic nature of saponin biosynthesis and its modulation by genetic and environmental factors [126]. This underscores the nutritional significance of sprouted soybean as a functional food, offering enhanced health benefits due to its increased saponin content (Figure 4).

#### 3.1.5. Carbohydrates

It was indicated that soybean seeds contain about 33% carbohydrates, of which soluble sugar stands at 16.6%. The major soluble sugars in the fruit pulp are sucrose, raffinose, and stachyose, with glucose and fructose in small quantities [127]. In more detail, the content of the soluble sugar shall be at a maximum of 90%, and consequently, sucrose shall constitute 41.3–67.5% of the specific weight, raffinose—5.2–15.8%, and stachyose—12.1–35.2%. Kjeld Sorensen named glucose, fructose, and sucrose welfare sugars because of their flavors and easy breakdown. Raffinose and stachyose are regarded as anti-nutritional factors, mainly because they do not break down in the human small intestine and can cause gastrointestinal disturbances such as flatulence and diarrhea [128]. However, data on the composition of sugars in soybean seeds and fluctuations within this composition depend on genetic qualities and environmental factors. The total sugars also show a very significant drop throughout the germination process. According to the research, the percentage of sugar decreases from 19.9% in the dry seeds to 14% in the 7-day sprouted seeds. Furthermore, changes in the germination of some soybean cultivars show a significant reduction of more than 90 percent in raffinose and stachyose content [129]. This selective degradation of anti-nutritional sugars improves the nutritional quality of the soybean sprouts because the relative proportion of dietary sugars is higher than those of harmful edible sprouts, thus enhancing their digestibility and acceptability. In general, soybean sprouting provides a valuable strategy for improving the nutritional quality of soybeans by eliminating deleterious sugars and increasing beneficial sugars for good digestive tract health [130]. Such preparation makes sprouted soybeans easy to digest and nutritious; thus, they can be recommended for those who want to include soy products in their diets (Figure 4).

#### 3.1.6. Essential Minerals

Soybean sprouts also contain essential minerals needed in the human diet; these include zinc, sodium, iron, calcium, and the rest. These minerals are not uniformly distributed in different concentrations in hypocotyls and cotyledons of the sprouts [131]. It is remarkable that the process of sprouting leads to changes in the mineral content of soybeans in the observed experiments. A literature review has shown appreciable changes in mineral content during sprouting, as demonstrated by comparative studies. In this case, calcium, sodium, manganese, potassium, and copper concentrations are higher, while the iron content is lower [132]. For instance, iron content decreases from 48.87 mg/g dry weight in raw seeds to 35.29 mg/g dry weight in 4-day-old sprouts. This shift in mineral composition is attributed to the sprouting process. However, it is essential to note that mineral composition in soybean sprouts can vary depending on factors such as cultivar and sprouting conditions [133]. Despite these variations, soybean sprouts remain a nutrient-dense food option. The enhanced mineral profile of soybean sprouts supports various physiological functions, including bone health, energy production, immune function, and heart health. Incorporating soybean sprouts into one’s diet can provide essential minerals, promoting overall well-being [134]. The nutritional benefits of soybean sprouts make them an excellent addition to a balanced diet, particularly for individuals seeking plant-based sources of essential minerals.

The reference analyzed the organ-specific accumulation of metabolites, phytohormones, and gene expression in dehydration-treated soybeans. Raffinose, trehalose, and cis-zeatin (cZ) accumulated specifically in roots under dehydration conditions [135]. Raffinose and trehalose likely play roles in osmotic adjustment, while cZ may contribute to root elongation for enhanced water uptake. The levels of raffinose, trehalose, and galactinol were correlated with the expression of key biosynthetic enzymes in individual plants but not at the organ level [136]. These enzymes were primarily expressed in aerial plants, suggesting that these metabolites are synthesized in the shoots and transported to the roots. Additionally, cZ and abscisic acid (ABA) levels were correlated with gene expression involved in their biosynthesis and metabolism at the plant level but not at the organ level [136]. This suggests that the transport process is crucial for organ-specific production of ABA and cZ during dehydration. This study underscores the importance of metabolites and phytohormone transport in mediating the organ-specific response of soybeans to dehydration (Figure 4).

### 3.2. Metabolomics and Nutrient Analysis

Recently, metabolomics has found its way into crop research as it helps determine crops’ quality and nutritional value. This is made possible by a modern metabolomics database and computational platform through which large datasets of metabolite information can be comprehensively evaluated and dissected. For instance, assessing the nutritional efficiency of genetically modified (GM) crops often requires advanced metabolomics techniques that go beyond conventional methods [137]. An example of the above is the glyphosate-tolerant GM soybean that has a glyphosate-resistant EPSPS enzyme from the gene of Agrobacterium tumefaciens strain CP4 incorporated within the plant to enable it to withstand being sprayed with glyphosate. This modification allows soybeans to withstand glyphosate application, which affects the EPSPS enzyme required for the pathway synthesis of aromatic amino acids. Using HPLC and chemometrics methods such as PCA, researchers determined amino acid content in glyphosate-resistant soybean. These tools effectively differentiate metabolic responses based on genetic variations, offering valuable insights for breeding and trait selection [138]. Similarly, another study applied metabolomics analysis using GC-MS and LC-MS to investigate the impact of CRISPR/Cas9-mediated editing of the β-ketoacyl-[acyl carrier protein] synthase 1 (KASI) gene in soybeans. The analysis revealed an 8–10% increase in seed sucrose content and a corresponding decrease in seed oil content, demonstrating the utility of metabolomics in uncovering the metabolic consequences of genetic modifications and providing insights into the role of KASI in soybean fatty acid biosynthesis and nutrient regulation [139]. Similarly, *CRISPR/ Cas9*-mediated editing of the *AIP2* gene, combined with biolistic transformation, enhanced soybean seed protein content without compromising oil content, as conformed through metabolomics promising using computational tools [140]. Sim and colleagues identified and quantified 266 putative metabolites, including carbohydrates, amino acids, lipids, and secondary metabolites, showing a more significant reduction in metabolite abundance under heat than drought stress [64]. In conclusion, metabolomics combined with computational tools enabled detailed profiling of metabolites, which can be used to assess the impact of genetic modification and environmental factors on crop quality and nutritional value (Figure 5).

### 3.3. Health Implications

Amino acids and proteins serve as fundamental building blocks of life. In humans, amino acids are utilized by every cell to synthesize proteins, which play critical roles in various physiological processes, including nutrient transport, nutrient storage, and cellular structure development. Soybeans are a vital source of phytosterols, essential for producing steroid hormones [141]. They have been processed into various food products, including soy oil, tofu, soy milk, soy sauce, soy flour, textured soy protein, soy protein concentrate, isolates, and soy-based infant formulas. These products are staples in Asian diets and have gained global popularity due to their versatility, taste, nutritional value, and environmental benefits. Soy foods are widely recognized for their numerous health benefits and are linked to a reduced risk of certain diseases [142]. Research suggests that soy consumption supports overall health by addressing common nutritional deficiencies often observed in vegetarians, such as low calcium, vitamin D, vitamin B12, and iron intake [143]. Furthermore, soy foods offer functional benefits, including lowering high cholesterol and blood pressure, preventing cardiovascular diseases, and managing conditions like type 2 diabetes, asthma, and osteoporosis. The health benefits of soy extend to various aspects of well-being [114]. Soy has been shown to slow the progression of kidney diseases, reduce the risk of lung, endometrial, prostate, and thyroid cancers, and improve lung function and memory. In women’s health, soy addresses several concerns, including breast pain, hot flashes, menopausal symptoms, premenstrual syndrome (PMS), and preventing breast cancer [144]. Additionally, soy aids in treating common ailments such as constipation, diarrhea, and muscle soreness caused by exercise. It also reduces protein levels in urine for individuals with kidney diseases. The versatility and nutritional value of soy products make them an excellent addition to a balanced diet [145]. As soy products’ global familiarity and acceptance continue to grow, their status as a popular choice for health-conscious individuals is solidified. Offering both preventive and therapeutic benefits across a wide range of conditions, soy products have become an integral part of a healthy lifestyle (Figure 5).

## 4. Computational Approaches in Soybean Disease Detection and Management

Plant diseases represent significant challenges to global agricultural production, with the Food and Agriculture Organization (FAO) estimating that pests and diseases contribute to a 29–20% reduction in crop yields [146]. Early detection of these diseases is critical to minimizing their impact; however, traditional manual monitoring methods often fall short of achieving timely diagnosis. Integrating Internet of Things (IoT) devices, sensors, drones, artificial intelligence (AI), and blockchain technologies is transforming agricultural practices, enhancing disease management and overall farm efficiency [147]. IoT sensors collect real-time data on soil moisture, fertilizer levels, weather conditions, and machine performance, which supports data-driven decision-making when coupled with AI [148]. The advancement in digital image processing, mainly through integrating deep learning (DL) algorithms, has significantly improved crop detection and disease diagnosis [149]. These DL techniques, particularly convolutional neural networks (CNNs), K-nearest neighbors (KNN) support vector machines (SVM), and artificial neural networks (ANNs), have demonstrated superior performance over traditional methods in classifying and diagnosing crop diseases [150]. Studies indicate that vision transformer (ViT) coupled with CNN models are highly effective for classifying, detecting, and segmenting plant diseases. Applying these models to image data, such as images of disease plants, has enabled early detection and accurate classification, which is crucial for managing crop health [151]. Soybean Seed Defect Identification Network (SSDINet) is proposed for the rapid and precise classification of defective seeds in soybean seed quality control. SSDINet is a lightweight network incorporating CNN, depth Wies separable convolutions, and squeeze-and-excitation blocks to optimize performance [152]. The results indicate that SSDINet outperforms existing models with an accuracy of 98.64%, using only 1.15 million parameters and achieving a processing time of 4.70 ms, demonstrating high efficiency and accuracy [153]. This article also focuses on the state-of-the-art machine learning (ML) and DL approaches used for crop disease diagnosis, emphasizing their performance and potential in soybean management (Figure 6).

### 4.1. Disease Identification Through Deep Learning

Soybean diseases have been a subject of interest in terms of diagnosis, wherein various methods such as digital image processing, pattern recognition, and computer vision have been applied by researchers to overcome this challenge [153]. This research introduces a learning methodology that utilizes AlexNet and GoogleNet convolutional neural network (CNN) architectures to develop a classification model for one non-disease category and three soybean disease classes: diseases include bacterial blight, brown spot, and frogeye leaf spot (FLS) [154]. The classification accuracies achieved by the AlexNet and GoogleNet models were 98.75% and 96.25%, respectively. The classification process involved fine-tuning various hyperparameters, including mini-batch size, maximum epoch, and bias learning rates for both CNN models [155]. The experimental outcomes demonstrate that the proposed deep CNN model significantly outperformed traditional machine learning approaches in soybean disease identification.

With the advancement of precision agriculture technologies, significant efforts have been directed toward classifying plant diseases. However, the performance of existing methodologies remains suboptimal. Additionally, many prior studies fail to accurately segment the leaf region from the entire image, particularly in cases with a complex background [156]. To address these challenges, a novel computer vision-based approach comprising two key modules is proposed. The first module categorizes the leaf by isolating the leaf region from the complex background. The second module introduces a deep learning-based convolutional neural network (CNN), SoyNet, for classifying soybean plant diseases using segmented leaf images [156]. Experiments were conducted on the “Image Database of Plant Disease Symptoms”, which comprised 16 categories. The proposed model achieved a classification accuracy of 98.14% and superior precision, recall, and F1 score [157]. Comparative analysis was performed against three state-of-the-art methods utilizing handcraft features and six widely used CNN architectures, including VGG19, Google LeNet, Dense Net 121, Xception Net, LeNet, and ResNet50 [158,159]. The results demonstrate that the proposed methods outperform these nine state-of-the-art techniques. This study primarily considers three categories of soybean leaves: healthy, infected, and unknown. The “infected” category includes 14 sub-types, resulting in 16 categories analyzed in this work. The sonnet model with a 70–30% train–test split achieved an accuracy of 98.14% and high precision, recall, and f1-score when the train–test set was 70–30% [160]. This study demonstrates the proposed method’s efficacy compared to three handcrafted state-of-the-art techniques and six deep convolutional neural network (CNN) models. The result indicates that high accuracy can be achieved by enhancing the diversity of pooling strategies, incorporating an ReLU activation function alongside dropout regularization, and performing multiple parameter tuning adjustments to optimize model performance [161]. Krishnaswamy’s AlexNet and VGG16 CNN models identified tomato diseases with 97.49% and 97.29% accuracy, respectively [162]. Furthermore, CNN-based transfer learning has been applied to identify soybean disease, including soybean rust (caused by *Phakopsora pachyrhizi*), frogeye leaf spot (caused by *Cercospora sojina*), and downy mildew (caused by Peronospora manshurica) [162]. The study makes two primary contributions: applying pre-trained AlexNet and GoogleNet CNN models on a large dataset to enhance classification accuracy and precisely identify disease symptoms in infected soybean leaves, providing valuable diagnostic support for plant pathologists [163]. These efforts demonstrate deep learning’s transformative role in agricultural disease management, enabling scalable and precise diagnostic solutions. Furthermore, another study developed a CNN architecture for plant identification based on leaf images consisting of five layers with ReLU or ELU activation function and MaxPooling applied to each pooling layer [164]. The system was tested on two leaf datasets: Flavia (1907 sample, 32 species) and Swedish (1125 samples, 15 species) using 160 × 160 pixel grayscale images of single leaves against a uniform background. The model achieved classification accuracies of 97.24% and 99.11% for the respective datasets, demonstrating competitive performance compared to recent methods for leaf feature extraction and classification [165].

A deep learning approach with a five-layer convolutional network and fully connected layers trained on 1.8 million images from the ILSVRC 2012 dataset achieved an average precision of 0.486 for plant identification tasks [166]. Preexisting deep CNN architectures such as AlexNet and GoogleNet were used to classify plant disease using a dataset of 54,306 images achieving 99.35% accuracy for healthy and different environments while highlighting the potential of deep CNNs for plant disease classification, though with limitations in cross-environment generalization [167]. A CNN-based system recognized ten rice diseases with an accuracy of 95.45% [168]. A pre-trained AlexNet model achieved a remarkable 99.35% classification accuracy for 26 diseases across 14 crop species using a large dataset of 54,306 images [169]. A CNN model for detecting and classifying plant diseases from simple leaf images achieved an impressive accuracy of 99.53% [170]. Similarly, CNN models such as AlexNet and VGG16 identified tomato diseases with accuracies of 97.29% and 97.49%, respectively [171]. A VGG-based model trained on the extensive PlantVillage dataset obtained 93.5% accuracy for multispecies plant identification [172]. A Capsule Network (CapsNet) model classified tomato disease from the Kaggle Plant Diseases database, achieving a high accuracy of 96.39% with focusing on large-scale disease detection [173]. The MobileNetV2 model achieved 99.36% accuracy on a smaller dataset to identify apple images from the PlantVillage database [174]. A DNN-based model classified mangoes in a small-scale dataset with 98.57% accuracy, demonstrating effective object detection capabilities [175]. The PlaNet model classified various plant species across multiple datasets, achieving 97.95% accuracy on large-scale datasets [176] (Figure 6). Another study applied AlexNet and GoogleNet to classify soybean leaves into four categories, one healthy and three diseased classes: bacterial blight, brown spot, and FLS, showing 98.75% and 96.25% accuracy, respectively [177]. Future improvements could focus on further fine-tuning hyperparameters such as mini-batch size and learning rates to enhance performance even more. In another study, the Soynet CNN for soybean disease recognition, a new CNN model was proposed specifically designed for recognizing soybean plant diseases [178]. This approach involved two key modules: background subtraction, which helps extract the soybean leaf from the image, and the Soynet model, which processes the images to detect diseases. Soynet was tested on the image database, containing sixteen disease categories containing plant disease symptoms. It achieved an impressive 98.14% accuracy, demonstrating high precision, excellent recall, and an F1 score [179]. When compared to three handcrafted feature-based methods and six state-of-the-art CNN architectures, including VGG9, GoogleNet, DenseNet121, Efficient Net, ResNet50, and LeNet, Soynet outperformed all of them, showcasing its superior effectiveness in plant disease classification [180].

### 4.2. Model Performance and Limitations

According to Google Trent data, the term “deep learning” has experienced a significant upward trajectory in search interest since 2013. The rise in popularity continued through 2016 and beyond, highlighting the widespread adoption of deep learning technologies in many fields [181]. The sustained interest peaking from 2022 to 2024 underscores the continued significance of deep learning in addressing complex problems and driving innovation across multiple disciplines [182]. Kunduracioglu and Pacal (2024) used both CNN and ViT models to classify grape images from the PlantVillage and Grapevine dataset, attaining perfect accuracy or 100% [183]. The ResNet-50 model was used to classify Arabica coffee leaves in the Plant Village dataset, achieving an accuracy of 98.54% on a small-scale dataset (Table 2) [184].

Deep learning has proven highly effective in diagnosing plant disease accurately. However, several challenges hinder its widespread application in this domain. One primary challenge is the scarcity of high-quality labeled data, which is both time-intensive and costly [185]. This data limitation can significantly impair the training of deep learning models’ performance. Another issue is a class imbalance, particularly in plant disease datasets, where certain disease classes are underrepresented. This imbalance can result in models performing well in majority classes but poorly in minority ones, compromising their generalization ability [186]. Factors such as lighting conditions, disease stage, and image resolution can affect the accuracy of the model’s predictions. Overlifting is another common problem where the model memorizes training data rather than learning generalizable patterns. While an overfitted model may perform well on training data, it often fails to generalize effectively to unseen data [186]. Most existing studies rely on a single dataset, which may limit the model’s accuracy when applied to real-world scenarios, as these datasets may not capture the full diversity of plant disease and environmental conditions. A more diverse and realistic dataset is needed to reflect better the variability encountered in practical applications. This will improve the robustness of deep learning models and ensure better performance in real-world disease diagnosis.
plants-14-00671-t002_Table 2Table 2Overview of various deep learning models and techniques.Model/StudyDisease(s) IdentifiedTechniques AccuracyKey FindingsLimitationsReferenceSoybean Seed Defect Identification Network (SSDINet)Soybean Seed DefectsCNN, Depth wise separable convolutions, Squeeze-and-excitation blocks98.64%Lightweight network with 1.15 million parameters and a processing time of 4.70 ms.It may not generalize well across different crops or disease types.[152]AlexNet and GoogleNet for Soybean DiseasesBacterial blight, Brown spot, FLSConvolutional Neural Networks (CNNs), Hyperparameter tuningAlexNet: 98.75%, GoogleNet: 96.25%The deep CNN model outperformed traditional machine learning methods in disease identification.Limited to specific soybean diseases, may struggle with complex backgrounds.[154,155]SoyNetSoybean diseases (16 categories)CNN, Background subtraction, Deep learning-based model98.14%Outperformed nine other CNN models, including VGG19, ResNet50, and LeNet.Background subtraction may not work well in noisy or cluttered images.[160]Transfer Learning (AlexNet and GoogleNet)Soybean diseases (3 classes)Transfer learning, Pre-trained CNN modelsAlexNet: 98.75%, GoogleNet: 96.25%Achieved high classification accuracy for disease identification using leaf images.Transfer learning may not always generalize well to novel disease types.[163,187]Soybean Disease Recognition ModelBacterial blight, Brown spot, FLSCNN, Background subtraction, SoyNet model98.14%Superior accuracy, precision, recall, and F1-score for disease recognition.May face challenges with complex backgrounds and diverse lighting conditions.[179]Tomato DiseaseTomato diseasesAlexNet, VGG16, CNNsAlexNet: 97.49%, VGG16: 97.29%Achieved high accuracy in identifying tomato diseases using CNN models.The dataset size is limited and may not perform well with rare tomato diseases.[162]Plant DiseaseHealthy and various environmental conditionsCNN (AlexNet, GoogleNet)99.35%CNN achieved high accuracy across different environments but faced challenges in generalizing.Limited cross-environment generalization may struggle in extreme weather conditions.[167]Rice DiseaseRice diseasesCNNs95.45%High accuracy for detecting rice diseases using CNN models.It may not generalize well to other crops beyond rice.[168]Tomato DiseaseTomato diseases (26 types)Pre-trained CNN (AlexNet)99.35%Remarkable accuracy for identifying 26 diseases in 14 crop species using a large dataset.It may overfit certain disease types and lack robustness in real-world field conditions.[169]Tomato DiseaseTomato diseasesCapsule Network (CapsNet)96.39%Focused on large-scale tomato disease detection with high classification accuracy.Requires large datasets for training and may not generalize well to small datasets.[173]Apple DiseaseApple diseasesMobileNetV299.36%Achieved exceptional accuracy using a smaller dataset for apple disease classification.It may not perform as well on larger, more diverse datasets.[174]Mango ClassificationMango classificationDNN-based model98.57%Effective object detection for mangoes with high classification accuracy.Limited to small-scale datasets, it may not be generalized to large-scale deployments.[175]Plant SpeciesMultiple plant speciesPlaNet model97.95%High accuracy was achieved for large-scale plant species classification.It may not handle rare species well; dataset dependency is critical.[176]


Table 2 overviews various deep learning models and techniques applied to plant disease detection, including their key findings, accuracy, and limitations. It highlights the effectiveness of CNNs, transfer learning, and other methods in identifying and classifying plant diseases across different crops.

### 4.3. Future Directions

IoT sensors deployed across agricultural fields can continuously collect data on environmental conditions such as temperature, humidity, soil moisture, and light intensity. Drones equipped with high-resolution cameras or multispectral sensors can further enhance monitoring by capturing detailed images of crops, enabling early disease detection before symptoms become visible to the naked eye [188]. By combining data from IoT sensors, drone images such as CNNs and ViTs can be used to analyze and detect potential diseases. For instance, abnormal changes in environmental conditions like fluctuating temperature or moisture could indicate the onset of diseases like fungal infections or bacterial growth. Real-time alerts can be sent through mobile apps, SMS, or other communication platforms, notifying farmers immediately when disease outbreaks are likely. These alerts can also consist of recommendations for applying fungicides or maintaining irrigation based on the type of disease and its severity. All these alerts can also be linked to automated crop treatment systems such as drones or robotic sprayers, which can take action based on disease detection without manual intervention. Additionally, cloud-based platforms can analyze the data from IoT devices, offering real-time insight into trend forecasts, which helps farmers make decisions for their crops [189]. This integration of IoT-based disease detection, automated treatment, and predictive analytics provides a comprehensive early warning system that enhances the efficiency, sustainability, and profitability of modern farming (Figure 7).

## 5. Predictive Modeling for Soybean Yield Optimization

### Machine Learning in Yield Prediction

Accurate prediction of soybean yield is crucial for agricultural production, monitoring, and early warning systems. Advancements in farming technology have significantly enhanced various aspects of the industry, including crop yields, crop selection, and resource management. Precision agriculture, which leverages technologies like GPS mapping, remote sensing, drones, and machine learning (ML), aims to optimize farming practices by collecting real-time data on environmental factors such as weather, soil properties, and crop growth [190]. These data are analyzed to determine optimal planting schedules, fertilizer and water requirements, and ideal harvest times, ultimately improving productivity and sustainability while reducing waste and environmental impact. Crop yield prediction, in particular, is a challenging task, and several ML models have been developed to enhance accuracy [191]. For example, a hybrid model combining multiple linear regression (MLR) and artificial neural networks (ANN) has been proposed to improve yield field prediction, achieving a high accuracy (R^2^ = 0.997). This study utilized daily meteorological data and soybean yield data from 173 country-level regions and meteorological stations across two major soybean-producing areas in China (Northeast China and Huang-Huai region), spanning 34 years [192]. Three machine learning algorithms, K-nearest neighbor, random forest, and support vector regression, were selected as base models to build a highly accurate and reliable soybean meteorological yield prediction model using a stacking ensemble learning framework [193]. To enhance the model generalizability, 5-fold cross-validation was applied, and optimization was performed through principal component analysis and hyperparameter tuning. The model accuracy was evaluated through five years of sliding prediction and four regression indicators across the 173 counties. Results showed that the stacking model outperformed other models’ accuracy and robustness, with the mean absolute percentage error (MAPE) being less than 5% [158]. The stacking prediction model effectively captured the spatiotemporal distribution of soybean yield, providing a new, more accurate approach to forecasting soybean yields.

This study developed climate-based soybean yield prediction models using machine learning (ML) to identify optimal planting sites in Mato Grosso do Sul. Meteorological data from 47 locations were used to calibrate algorithms, including MLR, MLP, SVM, RF, XGBOOSTING, and Grad BOOSTING. XGBOOSTING emerged as the most effective model with high accuracy and precision, achieving an R^2^ of 0.95 during calibration [194]. The study demonstrated significant spatial and seasonal variability in climate variables and their relationship with soybean yield. The models can aid in identifying high-yield zones and improving soybean farming practices. A study by Kaki and Wang found that environmental factors had a more significant influence on crop yield than genotype [195]. Random forest (RF) models have been shown to outperform other algorithms in crop yield prediction based on error analysis of different feature sets [196]. In addition, hybrid models combining machine learning with big data techniques, such as CNN and RNN frameworks, have been proposed to predict crop yield using historical data. Integrating remote sensing, weather data, and soil analysis has further advanced yield prediction models [197]. For instance, ML-based models that use satellite data for soybean yield forecasting in Brazil have significantly improved prediction accuracy. Hybrid models combining machine learning with crop modeling have also been explored to enhance yield forecasts in the US corn belt, achieving better prediction accuracy [198]. Other studies have incorporated decision trees, support vector machines (SVM), and genetic algorithms to create predictive models for crop yield and fertilizer recommendations [182]. Machine learning techniques have shown significant potential in improving crop yield prediction, disease detection, and precision farming, ultimately contributing to sustainable agricultural practices (Figure 7).

## 6. Economic and Environmental Viability of Soybeans as a Biofuel

### 6.1. Market Potential and Economic Analysis

The production of soybean biodiesel is a multi-stage process involving oil extraction, refining, and transesterification, which also generate high-value co-products like lecithin, tocopherols, squalene, and soybean meal [199]. These co-products contribute significantly to the economic viability of soybean biodiesel production, transforming it into an integrated biorefinery. Soybeans constitute 86% of raw material costs and 77% of total direct production costs, making soybean pricing a critical factor in cash flow and profitability [200,201]. Notably, 52% of revenue comes from soybean meal sales, with additional contributions from phospholipids (18%) and hydrocarbons (8%) (Figure 8) [202].

In 2020, the US produced approximately 11.57 million tons (MT) of soybean oil, of which 3.9 million MT were processed into soybean biodiesel and 1 million MT exported [203]. The increasing production of soybean biodiesel is expected to drive up the demand for and price of soybean oil while reducing the cost of soybean meal. This price decline occurs because more soybeans are diverted to oil extraction for biodiesel, thereby increasing the oil supply and decreasing the availability of soy meal, which could lower feed costs for farmers [204]. Energy industry analysts forecast a substantial increase in domestic biodiesel production, expecting output to quadruple from 550 million gallons in 2020 to 2 billion by 2022. To meet this expansion, U.S. feedstock producers must significantly scale up capacity, adding millions of metric tons of soybean oil to their annual production volumes [205]. An analysis of 11 different biofuel feedstocks concluded that none are economically viable, even at a USD 50 per barrel oil price. Various biomass-based biodiesel exhibited the lowest costs, while rapeseed biodiesel had the highest positive overall government budget impact [206]. Biodiesel production has surged in Brazil, from 736 million liters in 2021 to 5.42 billion liters by 2024, with soybean oil accounting for over 70% of the feedstock used in biodiesel production [207]. While soybean biodiesel has emerged as a key domestic fuel source, reducing reliance on fossil diesel and contributing to environmental sustainability, challenges remain. Feedstock availability, volatile soybean prices, and policy constraints in key regions, such as the U.S. and Brazil, affect the industry’s growth [208,209]. Comparative studies on biofuel feedstock have shown that biomass-based biodiesel, including soybean biodiesel, offers the lowest production cost. However, financial viability is often reliant on subsidies on supportive policy [206].

### 6.2. Environmental Benefits and Sustainability

As global population growth drives both food demand and the need for biodiesel to reduce carbon emissions, soybean oil stands out as a leading biofuel. With a production share of 35%, soybean oil boasts a high oil yield of 5000 kg per hectare, substantial productivity of 4.28 tons per hectare, and a competitive market price of USD 660 per ton. The use of palm oil for biodiesel production has been found to offer environmental benefits, but there remains insufficient long-term data to assess its sustainability fully [210]. Palm oil production contributes significantly to greenhouse gas emissions, deforestation, and habitat destruction. A study adopted a multi-country approach to analyze biodiesel production and consumption in Sub-Saharan Africa, specifically in Botswana, Malawi, Mozambique, Namibia, South Africa, Tanzania, Zambia, and Zimbabwe [210]. The study aimed to identify the most cost-effective raw materials and foster regional cooperation in biodiesel production and by-product energy utilization. Another study concluded that while soybean oil, widely produced in regions like the Americas and Sub-Saharan Africa, is the lead expensive feedstock for biodiesel, jatropha oil production in Sub-Saharan Africa offers significantly higher job creation potential, generating five times more employment [210]. However, due to the high production costs associated with biodiesel, significant government support through subsidies or tax reductions is necessary. Sustainability, in this context, is not limited to the renewable nature of biodiesel but also includes social inclusion and the development of marginalized regions [210]. Further examination of the economic potential of jatropha biofuel in Botswana found the results unconvincing due to low yields attributable to the lack of prior breeding efforts and the unsuitability of degraded agricultural land for jatropha cultivation [211].

Further studies investigated the economic and environmental sustainability of blending Brassica carinata, a non-edible oilseed crop, with wheat and other plants as a second-generation biofuel source in Italy [212,213]. Their findings indicated that the yields and economic performance reported in the literature could only be realized under optimal conditions, suggesting that public subsidies are essential to ensure the financial viability of this approach. Regarding cost comparison, in February 2018, the price of palm oil in Indonesia was USD 655 per metric ton (USD 486 per euro ton) [213]. In contrast, the cost of Brassica carinata oil in Italy was USD 618 per euro ton, which is higher than the Indonesian price for palm oil. The overall crude oil price ranged between USD 525 and USD 598 per euro ton, further highlighting the cost disparity [213]. Additionally, other findings analyzed the economic viability of biodiesel production in Serbia. The country has the potential to produce between 128,000 and 266,000 tons of biodiesel from oilseed crops, with additional yields of 10,000 tons from waste cooking oil and 8000 tons from non-edible sources such as tomato, grape, and tobacco seeds [213].

However, significant tax exemptions and government support can only realize this potential. The authors argue that using edible vegetable oils as a long-term biodiesel feedstock is unsustainable; therefore, alternative feedstock, particularly inedible oils and waste cooking oil, should be prioritized [214]. Centralized regulation of collection systems for waste cooking oil is essential to ensure a reliable supply. Pure biodiesel production in Bangladesh has been reported to be economically uncompetitive [214]. The estimated cost of biodiesel ranges from USD 1.60 to USD 2.396 per liter, compared to the USD 0.71 to USD 0.90 per liter paid for conventional diesel. A 20% mustard biodiesel blend costs approximately USD 0.777 per liter [215]. To make biodiesel production more cost-effective, the authors suggest reducing raw material costs, optimizing processing techniques, and recycling methanol used in transesterification, particularly if biodiesel production becomes commercially scaled [215]. An examination of the environmental and economic aspects of biodiesel production in Bulgaria from 2016 to 2020 found that the average price of biodiesel (B100) under environmental criteria was USD 428 per ton, 14% higher than the price based on economic factors (USD 378 per ton) [216]. However, when evaluated using environmental criteria, total greenhouse gas emissions were 6.6% lower, demonstrating the ecological advantages of biodiesel despite the higher costs associated with its production [216].

### 6.3. Modeling the Economic-Environmental Trade-Off

The integration of Life-Cycle Assessment (LCA) and Multi-Objective Optimization (MOO) has led to the development of Life-Cycle Multi-Objective (LCMO) frameworks, which are designed to assess and balance environmental and economic aspects of biodiesel production. LCA is a tool used to evaluate the ecological impacts of different feedstocks, while MOO helps optimize the biodiesel blend by considering multiple objectives simultaneously [217]. One such approach is the Life-Cycle Multi-Objective Chance Constrained (LCMO-CC) model, which aims to determine optimal biodiesel blends by minimizing costs and environmental impacts while accounting for the uncertainty in feedstock composition, which consists of a combination of vegetable oils, animal fats, and waste cooking oils, with specific proportions varying based on availability and optimization criteria [217].

The model considers various feedstocks, such as palm, rapeseed, soybean, and waste cooking oil (WCO), with their associated uncertainties in chemical composition, costs, and environmental impacts. The model output is an optimal blend of feedstocks that meet biodiesel specifications (e.g., density, cetane number, cold filter plugging point) at minimal cost and environmental impact [217]. Due to the trade-offs between cost and environmental impacts, the model helps decision-makers identify Pareto-optimal solutions, where no improvement in one objective can be made without worsening another [217]. The biodiesel production cost is primarily driven by feedstock prices, which typically account for approximately 85% of the total cost [218]. Prices for feedstocks from 2011 to May 2014 were sourced from IndexMundi (2014) and Grennea (2014), with WCO prices being notably closer to those of virgin oils in July 2013, representing a conservative scenario for evaluating the benefits of WCO [218]. Given the availability of detailed data on biodiesel production in the country, the Portuguese context was chosen as a case study. However, the model can be replicated for biodiesel production in other regions [219].

The LCA model further evaluates the environmental impacts of four feedstocks, palm, soybean, rapeseed, and WCO, using data from multiple sources, including Ecoinvent databases and various studies [219]. The assessment includes impacts on GHG emissions, water consumption, and water quality degradation (via impact categories FE, AC, HT, and ET). The system boundaries for crop-based oil systems encompass cultivation, oil extraction, transportation, and refining, with cultivation sites in countries such as Colombia, Malaysia, Argentina, Brazil, and the US. In contrast, the WCO supply chain focuses on collection and refining within Portugal, with differing processes for low- and high-quality WCO based on free fatty acid content [219]. To address feedstock compositional uncertainty, the model employs Chance Constrained Programming (CCP) optimization, a technique used in blend optimization to minimize risks associated with feedstock price volatility and GHG emissions variability. CCP has been applied to conventional feedstocks and secondary materials like WCO, showing that blending can help control costs while ensuring biodiesel fuel quality. For technical compliance, the blend must meet specific biodiesel properties (density, cetane number, cold filter plugging point, iodine value, and oxidative stability) with constraints based on existing prediction models [220]. The mathematical formulation of the optimization problem is aimed at determining the Pareto-optimal blend by minimizing both production costs and environmental impacts. This is completed by calculating the total effect for each feedstock based on its price and ecological coefficients [220]. Constraints are imposed to ensure compliance with technical specifications and the distribution of risks based on a chosen tolerance level. The demand was set to 1, and no supply limitations were considered, as the primary focus was determining the feedstocks’ optimal blend ratio [220].

Sustainability in biodiesel production involves economic and environmental aspects and social considerations [217]. Economic sustainability emphasizes competitiveness with other energy sources, while social sustainability addresses equitable access to resources like food and health [217]. Environmental sustainability focuses on impacts such as soil and water quality, GHG emissions, and biodiversity. A holistic sustainability assessment requires the consideration of not just the primary biodiesel product but also co-products like glycerol, which can play a critical role in the economic viability of biorefineries [221]. The main driver for biodiesel expansion will be cost-effectiveness, achieved using cheaper and non-edible raw materials and more energy-efficient technologies supported by government policies [213]. Furthermore, due to global water scarcity, the water footprint is becoming an increasingly important consideration in biodiesel production. For instance, producing one liter of biodiesel from rapeseed or soybean requires approximately 14,000 L of water [222]. Thus, future sustainability assessments should integrate both environmental and social dimensions, considering the primary fuel product and its co-products, to ensure biodiesel’s long-term viability and acceptance as a renewable energy source. In addition to the water footprint, the carbon footprint of soybean-based biodiesel production is a critical factor in assessing its environmental sustainability. Life cycle analysis has shown that using pure biodiesel (B100) can reduce carbon dioxide emission by approximately 74% compared to petroleum diesel [223]. Studies indicated that soybean biodiesel reduces greenhouse gas (GHG) emissions by approximately 40% to 69% compared to fossil diesel, considering emissions from cultivation processing and combustion [224]. Additionally, the adaptation of precision agriculture, reduced nitrogen fertilizer use, and renewable energy in processing facilities can significantly reduce the carbon footprint of soybean biodiesel.

### 6.4. Ecological Impact of Soybean Production for Biofuels

Soybean-based biodiesel is renewable but poses huge environmental concerns. Studies show that soybean biodiesel emits fewer CO_2_ molecules than petroleum-based fuels, yet its carbon footprint is influenced by emissions from land use changes and deforestation [225]. Large-scale soybean farming is a major deforestation driver, which has contributed to habitat loss and biodiversity loss in areas like the Amazon and Cerrado [226]. Resource depletion is heightened by water-intensive soybean farming, which takes about 9000 L of water for every liter of biodiesel produced [227]. Over-irrigation and chemical fertilizers contribute to nutrient depletion and degradation of the soil [228]. Furthermore, agrochemical runoff from soybean fields deteriorates water quality and alters microbial diversity [228]. Comparison studies indicate that soybean biodiesel has lower environmental impacts compared to fossil fuels but is not as sustainable as others, such as sugarcane ethanol and jatropha biodiesel [229]. Traditional monoculture agriculture increases soil erosion and pesticides, while sustainable techniques, such as no-till agriculture, crop rotation, and agroforestry, help counteract the harmful effects of conventional monoculture [230]. To this effect, challenges with sustainability, efforts such as the Round Table on Responsible Soy promote responsible agriculture with the least destruction of the environment [231].

## 7. Climate Change Impact on Soybean Growth and Productivity

### 7.1. Climate Variables Affecting Soybean Growth

Climate variables affecting soybean growth impact flowering time and maturity. As a highly adaptable crop, soybean thrives across a wide range of latitudes from 50° to 35°, making them a key species in global agriculture [232]. Such adaptability is primarily attributed to its ability to respond to environmental variables such as photoperiod, temperature, and abiotic stresses, which influence critical agronomic traits like flowering time and maturity [73]. Soybeans flowering is regulated by photoperiod sensitivity, with short days inducing flowering and long days delaying, which combines the effects of in-season temperature, governing the geographical adaptation of different soybean cultivars [73]. Soybean is classified as short-day plants, meaning their flowering and maturity are tightly linked to the length of daylight they receive [233]. The interplay between temperature and photoperiod regulates the timing of flowering and the plant’s overall growth cycle. These two factors are essential for determining the geographic range in which a soybean variety can be successfully cultivated (Figure 8) [233].

The literature data examines the impact of climate change on increased water stress during the critical reproductive development stages of soybeans (*Glycine max*), which is widely recognized as when soybeans are particularly susceptible to water limitations [234]. To assess these impacts, climate projections from three general circulation models (GCMs)—the Goddard Institute for Space Studies (GISS), the Geophysical Fluid Dynamics Laboratory (GFDL), and the United Kingdom Meteorological Office (UKMO)—were utilized to generate daily simulated weather datasets for nine climate centroids in Iowa [234]. The results revealed that water stress was most severe during the R2–R5 stages under the UKMO climate scenarios, and this stress intensified as the climate changed. In contrast, while water stress was observed under the GISS and GFDL scenarios, its severity did not increase with higher CO_2_ concentrations or changing climate conditions [234]. These findings suggest that the magnitude of water stress during the critical reproductive stages of soybeans will vary depending on the climate scenario, with the UKMO projections indicating a more substantial potential for future water limitations.

### 7.2. Computational Modeling of Climate Effects

Several studies have examined the impact of climate changes on soybean production using computational models to simulate crop response to changing environmental conditions. One approach to improve the accuracy of photosynthesis modeling in soybeans integrates a leaf-level coupled energy balance model, which considers photosynthesis, transpiration, and stomatal conductance. This method has been tested under controlled conditions, such as soil–plant–atmosphere–research (SPAR) chamber experiments and various temperature and CO_2_ concentration treatments [235]. Beyond photosynthesis modeling, statistical and combinational approaches have assessed the relationship between climate variables and soybean yield projection. Multiple regression analysis indicates a strong correlation between soybean yield, rainfall, and maximum temperature, explaining approximately 91.2% yield variability [236]. Future yield projections based on 150 climate models under four climate change scenarios suggest a 3.8% increase in average soybean yield. However, 30% of forecasts predict lower yield, while 70% indicate higher yield. This research emphasizes the uncertainty in the yield projection due to climate variability and the need for an improved productivity framework.

Multi-model simulation studies further highlight the uncertainty of soybean yield prediction under climate change. Comparative analysis of ten widely used crop models reveals significant variability in their yield responses to temperature and CO_2_ changes. For instance, under a +3C temperature rice, the model is estimated to range from a 24% decrease to a 29% increase in Argentina, while in Brazil, yield reduction varies from 38% to no significant changes [237]. Additionally, when CO_2_ levels increase from 360 ppm to 540 ppm, the projected soybean yield increase ranges from 6% to 31%, depending on the model used [237]. These discrepancies highlight the challenges in forecasting soybean sensitivity to temperature stress and CO_2_ responses.

Process-based soybean growth models, such as GLYCIM, SOYCIM, Root Zone Water Quality Model 2 (RZWQM2), and DSSAT CS_CROP GROW soybean, have been widely evaluated in climate impact studies. These model representations of soil–water–plant interaction differ and have been tested with experimental data from multiple soybean-growing regions [238,239]. Sensitivity analysis suggests that for every 100 ppm increase in CO_2_, soybean yield could rise by approximately 8.8%, while a 1 °C temperature increase may result in a 4.8% yield reduction [240]. However, model prediction is significant, with some estimating yield decline of 0.4% to 7.2%, while others indicated potential gain under specific climate conditions. Given these inconsistencies, researchers emphasize the need for a multi-model ensemble approach to enhance the reliability of soybean yield prediction [241,242]. Continued advances in model calibration and heat stress sensitivity remain essential for improving the accuracy of climate impact projection on soybean production (Figure 8).

## 8. Soybean Microbiome and Soil Health

### 8.1. Soybean-Specific Microbial Communities

The soybean rhizosphere harbors a diverse range of microbial communities that comprise nitrogen-fixing and nitrifying bacteria [243,244]. These microbial organisms play essential roles in soil fertility, plant growth, and stress adaptation [243,244]. Their associations are particularly relevant in saline environments, where maintaining nitrogen availability is crucial for soybean productivity [243,244]. Studies have confirmed that Bradyrhizobium inoculation helps in enhancing nifH gene expression; thereby, it aids in improving biological nitrogen fixation efficiency and altering diazotrophic community structures in soybean cultivation [245,246,247]. However, long-term nitrogen fertilization can significantly affect rhizosphere microbial diversity, which ultimately leads to shifts in nitrifying bacterial populations and potential imbalances in nitrogen cycling [248]. Nitrifying bacteria are essential for soybean nitrogen uptake as they convert ammonium (NH_4_^+^) into nitrate (NO_3_^−^), which plants can easily absorb. Research on soybean–maize intercropping has found a strong link between nitrification rates and nitrogen-use efficiency, indicating that these bacteria play a significant role in making nitrogen available in the rhizosphere [248]. Additionally, in saline soils, strategies that involve recruiting nitrifying bacteria and rhizosphere symbionts have been effective in helping soybeans adapt to salt stress [249]. A greenhouse experiment examined how soil sterilization and plastic film mulching affect soybean growth, emphasizing the role of microbial interactions. The findings showed that the US+F treatment (unsterilized soil with plastic film mulching) resulted in the best growth for soybeans, while the S+NF treatment (sterilized soil without plastic film mulching) led to the poorest outcomes [246]. Notably, plastic film mulching and soil microbiota contributed 56% and 10% to soybean growth, respectively, underscoring the importance of rhizosphere microbial communities in supporting plant development under stress conditions. Several microbial taxa that promote growth have been identified in soybean rhizospheres, especially those related to nitrification, nitrogen fixation, and salt tolerance. These include Methylovorus, Methylophilus, Oharaiibacter, Sudarthrobacter, Neobacillus, Peribacillus, Acinetobacter, Stenotrophomonas, and Neorizobium [244]. Furthermore, the Methylovorus OTU 3058 and Methylophilus OTU 3336, both from the Methylophilaceae family, show high salinity tolerance and aid in nitrate assimilation, which may help soybeans absorb nitrogen in degraded soils. Oharaiibacter OTU 3280, belonging to the Rhizobiales order, tolerates salinity levels ranging from 0 to 22% NaCl and has nitrogen-fixing capabilities, making it significant for legume symbiosis [250]. Additionally, Sudarthrobacter OTU 2779 produces indole acetic acid (IAA), a plant hormone that encourages root elongation and nutrient uptake, potentially improving soybean root structure in nitrogen-deficient soils [243]. Neobacillus OTU 3480 and Peribacillus OTU 369, recently reclassified from Bacillus spp., are beneficial nitrifying and nitrogen-cycling bacteria. Peribacillus supports maize germination under saline stress, while Neobacillus functions as a root endophyte, aiding nitrogen uptake in leguminous plants [248]. Stenotrophomonas OTU 3424, recognized as a plant growth-promoting bacterium (PGPB), produces IAA and exhibits osmotic stress tolerance, essential for soybean growth under variable soil nitrogen levels [245].

Fungal communities in the soybean rhizosphere also contribute to nitrogen cycling and soybean adaptation. Fungal genera such as Actiniales, Fusarium, and Pyrogemula are dominant in soybean rhizosphere genera that have been associated with improved nitrogen-use efficiency [244]. Additionally, the *Actiniales OTU 537*, an unidentified fungi found in soybean roots, has been associated with improved nitrogen-use efficiency [244]. Fusarium OTU 1525, despite being known as a plant pathogen, showed positive growth effects in soybeans, possibly due to secondary metabolites that strengthen plant resilience to abiotic stress [247]. The role of Pyrogemula OTU 130 in soybean nitrogen metabolism remains uncertain and requires further study [246]. In summary, different communities of bacteria and fungi have been identified in the soybean root rhizosphere, improving the growth and production of soybeans. However, further research is still required to validate their functional role to significantly enhance their application (Figure 9).

### 8.2. Computational Metagenomics in Soybean Rhizosphere

Metagenomics has become an essential method for exploring microbial diversity in soybean-cultivated soils. It helps identify key microbial taxa that influence soil health, nitrogen cycling, and plant productivity. Unlike traditional culture-based methods, metagenomics analyzes microbial DNA directly from environmental samples, offering a detailed view of rhizosphere and bulk soil communities [251]. This approach sheds light on microbial interactions in soybean root microbiomes, where beneficial microbes support nutrient mobilization, enhance stress tolerance, and promote biological nitrogen fixation. Soybean rhizosphere metagenomic studies have identified nitrogen-cycling bacteria that improve plant nitrogen uptake and soil fertility. Research on the soybean root nodule microbiome found Nitrobacter and other nitrogen-transforming bacteria, which play a crucial role in rhizosphere function and nitrogen assimilation [252]. Functional metagenomics has also shown that soybean cultivation alters soil microbial composition, affecting nutrient cycling, carbon sequestration, and overall soil health [253]. These findings emphasize the importance of rhizosphere microbial diversity for maintaining sustainable soil ecosystems in soybean agriculture. Computational metagenomic analyses have revealed shifts in microbial diversity across different soybean cropping systems. Long-term soybean cultivation has been associated with an increase in beneficial nitrogen-fixing bacteria and a decline in microbial taxa linked to pathogenic interactions, indicating enhanced microbial stability and disease resistance [254]. Studies on soybean intercropping systems have shown that cropping practices significantly affect rhizosphere prokaryotic communities, influencing soil microbial diversity and nitrogen uptake under stress conditions [255]. These findings demonstrate the impact of agricultural practices on the microbial ecology of soybean fields, soil fertility, and plant health. Metagenomic–metabolomic studies have also explored fermented soybean-derived microbiomes, identifying microbial consortia that support nutrient cycling and promote plant growth [256]. These discoveries highlight the importance of microbial interactions within the soybean rhizosphere, which play a key role in fostering a diverse microbiome essential for sustainable agriculture and improved nitrogen fixation efficiency.

The integration of metagenomics with computational bioinformatics tools has transformed soil microbiome research by identifying beneficial microbial consortia for sustainable agriculture [257]. High-throughput sequencing and functional gene analysis have enabled detailed profiling of microbial diversity, allowing researchers to understand the roles of root-associated bacterial communities in soybean soils. Future research should focus on advancing metagenomic applications in soybean cultivation to develop microbial-based strategies for enhancing nitrogen fixation, stress resilience, and soil health. These efforts are essential for sustainable soybean production, long-term soil fertility, and climate adaptation in agriculture.

### 8.3. Applications of the Soybean Microbiome for Sustainable Agriculture

The soybean microbiome is crucial for sustainable agriculture as it improves soil fertility, supports nitrogen fixation, strengthens crop resilience, and boosts productivity. Studies show that microbial communities linked to soybean roots and rhizosphere soil affect soil structure, nutrient cycling, and plant health. These microbes are key to lowering reliance on synthetic fertilizers and advancing agroecosystem sustainability [258]. Bradyrhizobium inoculants play a crucial role in biological nitrogen fixation by converting atmospheric nitrogen (N_2_) into bioavailable ammonia (NH_3_) for soybean uptake. Field studies indicate that inoculating with Bradyrhizobium significantly increases soybean biomass, grain yield, and soil microbial diversity [259]. Long-term use of these inoculants boosts beneficial bacterial populations, which aids in nutrient cycling and organic matter decomposition [260]. Research on soybean–alfalfa intercropping systems shows that soil pH and microbial composition affect rhizobial diversity and nitrogen-fixing efficiency, emphasizing the importance of microbial inoculants for sustainable legume cultivation [261]. Agricultural management practices have a significant effect on the soybean microbiome beyond nitrogen fixation. Practices such as crop rotation, conservation tillage, and the use of organic amendments improve soil microbial diversity, nutrient uptake efficiency, and plant health. Research indicates that long-term cultivation of glyphosate-resistant soybean alters soil microbial composition, but overall microbiome stability remains consistent across different cultivars [262]. Additionally, cover cropping and the application of organic fertilizers are associated with increased microbial diversity, enhanced nitrogen fixation, and better soil structure, which contribute to higher soybean yields in dryland production systems [263]. A study comparing high- and low-yield soybean field sites found notable differences in microbiome compositions. High-yield fields showed greater bacterial diversity and more enriched nitrogen-cycling communities, indicating that microbial diversity plays a crucial role in soybean productivity [264]. Furthermore, soybean root-associated microbiomes are affected by crop management strategies, which in turn influence nutrient acquisition and overall plant resilience [265]. Microbial contributions go beyond nitrogen fixation. The soybean rhizosphere microbiome is essential for nutrient mobilization, disease suppression, and tolerance to abiotic stress. Diverse bacterial communities in soybean soils are important for rhizospheric fertility, affecting microbial functions, plant physiology, and grain yield [260]. In soybean cropping systems, microbial interactions influence carbon and nitrogen cycling, which improves soil structure, organic matter decomposition, and nutrient exchange among microbes [266]. Integrating phytomicrobiome interactions into sustainable soybean agriculture has become a focus of research, as beneficial microbes enhance resource efficiency, plant–microbe signaling, and stress tolerance. Studies show that microbial communities in soybean systems improve drought resilience, phosphorus solubilization, and nitrogen assimilation, contributing to increased crop productivity [267]. These interactions facilitate the creation of “designer microbiomes” specifically designed to boost soybean seed yield and soil health across different environmental conditions. Advancements in metagenomics and microbial inoculation technologies are transforming sustainable soybean production. Research indicates that incorporating beneficial microbial communities into agricultural practices can significantly enhance soil microbiome functions, leading to reduced reliance on chemical fertilizers and improved soil restoration [268]. Studies on sustainable farming practices demonstrate that combining microbial inoculants with precision agriculture techniques can optimize soil health and boost long-term soybean productivity [269]. Utilizing the soybean microbiome through microbial community applications, sustainable crop management, and microbial biotechnology offers a promising approach to improving soil health, increasing crop resilience, and promoting sustainable soybean agriculture (Figure 9).

## 9. Future Perspectives in Integrative Soybean Research

### 9.1. Cross-Disciplinary Integration

Crop GPT is an innovative and collaborative strategy for soybean breeding that integrates an interdisciplinary approach within an Integrated Plant Data Bank (IPDP). It combines high-quality germplasm selection, batch cloning of functional genes through machine learning methods, high-throughput genotyping, phenotypic evaluation, and constructing an open platform for IPDP services [270]. Central to Crop GPT’s methodology is developing pre-trained AI encoders that process multi-modal soybean genomic data, continuously optimizing the system to deliver refined breeding suggestions. AI-driven techniques help facilitate the identification and cloning of functional genes or quantitative trait genes (QTGs), by leveraging gene networks and cutting-edge machine learning algorithms, making predictions about functional elements and gene domains, enabling gene editing technologies, such as knockouts, knock-ins, and single-base editing, to generate new alleles that could lead to novel phenotypes absent in natural germplasm [270]. AI models also predict regulatory relationships and upkeep genes, providing optimal combinations for soybean breeding and guiding the development of superior breeding materials. These materials can be integrated into soybean germplasm resources for future breeding cycles, enhancing genetic diversity and adaptability in Crop GPT. Machine learning and deep learning techniques are essential components of Crop GPT’s genetic prediction tools [270]. Machine learning methods, including support vector machines, random forests, and Light GBM, do not rely on prior assumptions and can capture non-linear relationships, making them highly effective for soybean genetic prediction [271]. While these methods require complex data pre-processing and are sensitive to noisy features, deep learning techniques excel at automatically learning feature representations, showing superior predictive performance across a range of soybean genomics tasks [271]. In genomic selection tasks, no significant differences have been observed between conventional statistical methods and modern machine learning/deep learning models, which can be attributed to factors such as non-linearity not being a characteristics of soybean genomics data, insufficient dataset size, or the fixed architecture of deep learning models not being optimal for the data at hand [271].

Genomic prediction is promising for accelerating selective soybean breeding using high-density markers across the genome. Unlike phenotype-based selection, genomic prediction captures all genetic variation and can conduct genetic evaluations without requiring phenotypic data, reducing breeding costs and shortening generation intervals [271]. Traditional genomic prediction methods, such as best linear unbiased prediction, GBLUP, and Bayesian models, including Bayes A, Bayes B, and Bayes Lasso, enhance soybean breeding efficiency by providing faster and more accurate forecasts of genetic traits [271]. In conclusion, Crop GPT represents a cutting-edge AI-driven solution for soybean breeding that optimizes genetic selection, accelerates breeding cycles, and integrates genomic prediction tools for developing elite soybean varieties. Its innovative use of AI and machine learning makes it a transformative approach in modern soybean agriculture, ensuring enhanced productivity and sustainability in soybean improvement programs.

### 9.2. Policy and Research Directions for Soybean Production

Soybean production is primarily driven by two main growth factors: the planted/harvested area and productivity. The planted area reflects farmers’ interest in soybean cultivation, which is influenced by community food needs and sociocultural conditions [232]. On the other hand, productivity is determined by applying production technologies, including the adoption of superior soybean varieties, and is further influenced by land suitability, climate, and technological adoption [272]. There are at least five key strategies to boost soybean production. One, improving market prices; two, optimizing the utilization of available land; three, intensifying and enhancing production processes; and five, ensuring programmed consistency and commitment from relevant stakeholders [272]. So far, the government has undertaken various initiatives, including developing new high-yield soybean varieties, to boost production. Adopting quality seeds derived from superior varieties is critical to improving yields, controlling pests, and mitigating adverse environmental conditions [273]. Farmers generally adopt superior soybean varieties more quickly than other production technologies, as they are simple to implement and cost-effective. However, achieving self-sufficiency in soybean production remains challenging, primarily due to the relatively low profitability of soybean farming compared to other crops [273].

The Ministry of Agriculture, through the Indonesian Legumes and Tuber Crops Research Institute, has developed a range of superior soybean varieties aimed at improving productivity at the farm level [274]. Despite these efforts, the adoption of these new varieties has been limited due to weaknesses in the seed distribution system and inadequate outreach efforts regarding the benefits of these varieties. The Ministry of Agriculture has developed several new soybean varieties with specific advantages, such as drought tolerance, shade resistance, and suitability for acid-prone soils. These include varieties such as *Demas 2014*, *Deena 2014*, *Deena 2 2014*, *Devon 2015*, *Daeger 2016*, *Daeger 2017*, *Daeger 2 2017*, *Deeven 2 2017*, *Detap 1 2017*, *Detap 2 2018*, *Derek 1 2018*, *Demus 2 2019*, *Demus 3 2019*, *Dering 2 2019*, and *Dering 3 2019* [274]. However, the dominant varieties in use by Indonesian farmers remain older varieties, such as Andresmoro, Grover, Willis, Argomullo, Bugrangrang, and Ballerin. Despite their higher yield potential, the lack of adoption of newer varieties is partly due to farmers’ preference for familiar varieties and the insufficient availability and quality of soybean seeds [274]. The availability of high-quality seeds at the dissemination site is essential for increasing adoption and productivity. Once local farmers have access to the new soybean varieties, they will likely share their knowledge with neighboring farmers, which can lead to broader adoption [275]. Adopting superior soybean varieties is also significantly influenced by factors such as access to irrigation, credit, and participation in farmer groups. Therefore, improving access to credit for soybean seed producers and providing market guarantees for seeds are critical government interventions to ensure the sustainability of soybean seed breeding efforts [276]. In conclusion, increasing the distribution and adoption of new soybean varieties requires addressing seed availability issues, improving outreach and extension services, and ensuring access to necessary agricultural resources such as irrigation and credit. These actions will support higher productivity and contribute to the national goal of reducing soybean imports and achieving self-sufficiency in soybean production.

## Figures and Tables

**Figure 1 plants-14-00671-f001:**
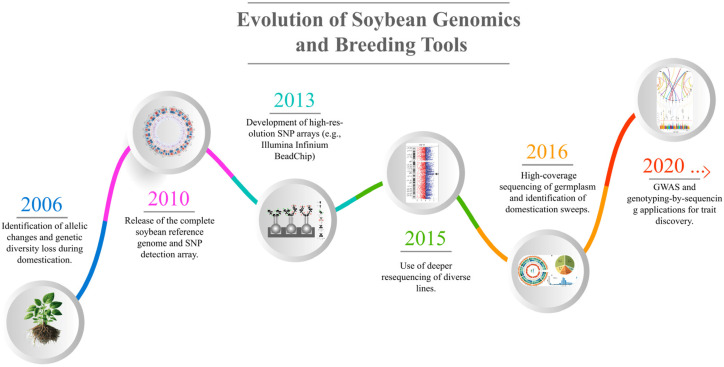
Illustrates key milestones in soybean genomics and breeding tools, starting with identifying genetic diversity loss in 2006, the release of the soybean reference genome in 2010, and the development of high-resolution SNP arrays in 2013. Advances continued with deeper resequencing in 2015, germplasm sequencing for domestication studies in 2016, and GWAS/genotyping-by-sequencing applications for trait discovery from 2020 onward.

**Figure 2 plants-14-00671-f002:**
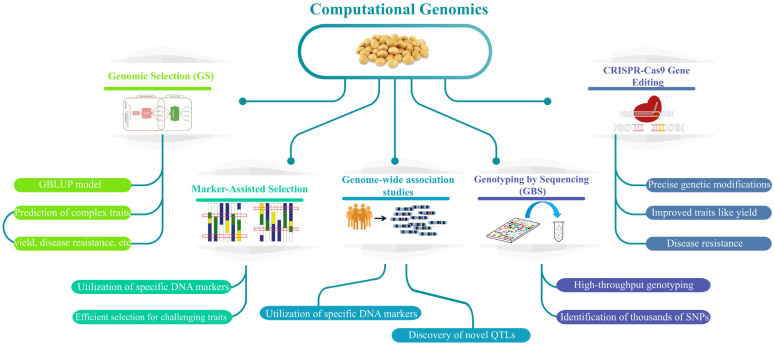
Highlights key computational genomics tools in soybean research, including genomic selection (GS) for predicting complex traits, marker-assisted selection for efficient DNA marker utilization, and genome-wide association studies for discovering novel QTLs. It also covers Genotyping by Sequencing (GBS) for high-throughput SNP identification and CRISPR-Cas9 gene editing for precise genetic modifications, improving traits like yield and disease resistance.

**Figure 3 plants-14-00671-f003:**
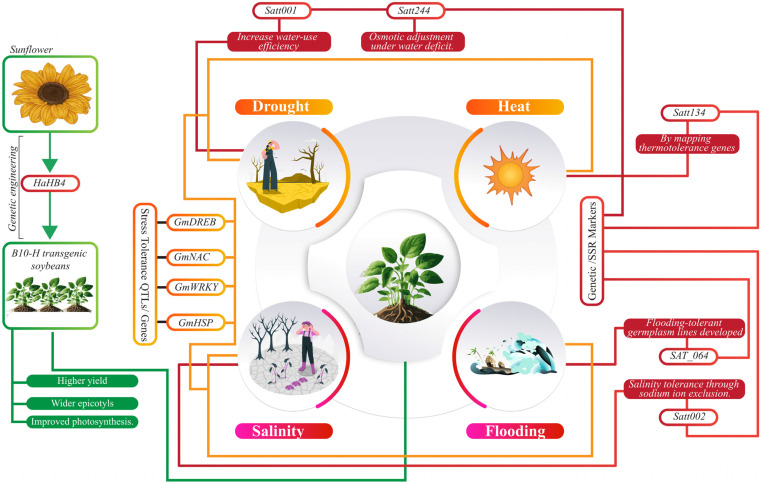
Illustrates key stress tolerance genes and plant markers categorized under drought, salinity, heat, and flooding. Genes such as HaHB4, GmDREB, and Satt134 enhance tolerance to drought, heat, and salinity through improved water-use efficiency, ion regulation, and thermotolerance. Genetic engineering and QTL/SSR markers play a crucial role in developing stress-resilient germplasm with enhanced yield and photosynthetic efficiency.

**Figure 4 plants-14-00671-f004:**
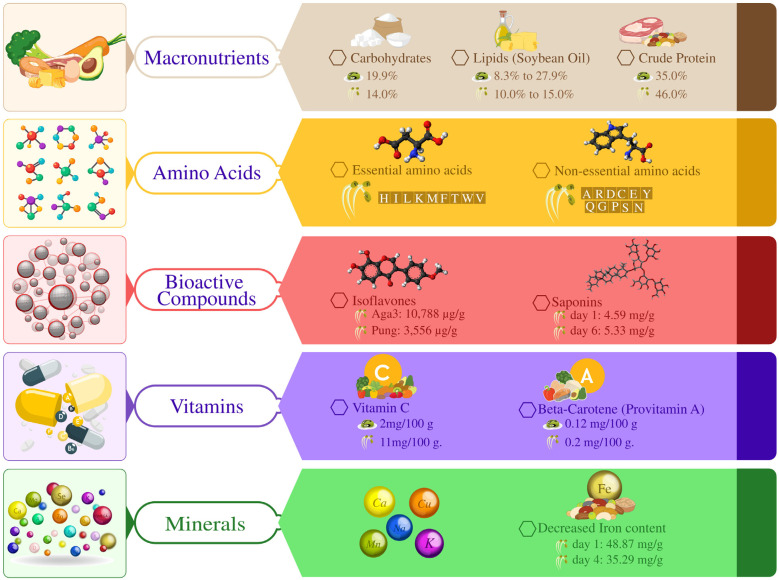
Summarizes the nutritional composition of soybeans, highlighting macronutrients (carbohydrates, lipids, and proteins), amino acids (essential and non-essential), and bioactive compounds like isoflavones and saponins. It also demonstrates key vitamins (C and beta-carotene) and minerals, including decreased iron content over time, emphasizing soybeans’ role in health and nutrition.

**Figure 5 plants-14-00671-f005:**
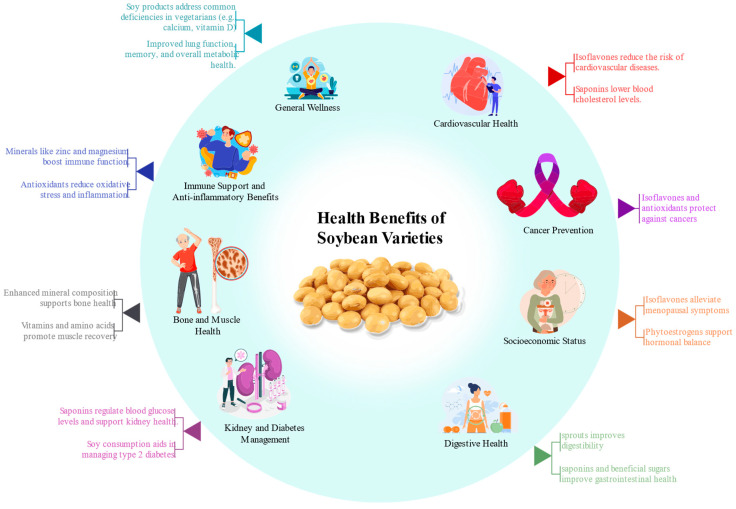
Highlights the health benefits of soybean varieties, including improving cardiovascular health, immune support, bone and muscle health, and managing diabetes. Soy compounds like isoflavones, saponins, and antioxidants play key roles in cancer prevention, hormonal balance, and digestive wellness.

**Figure 6 plants-14-00671-f006:**
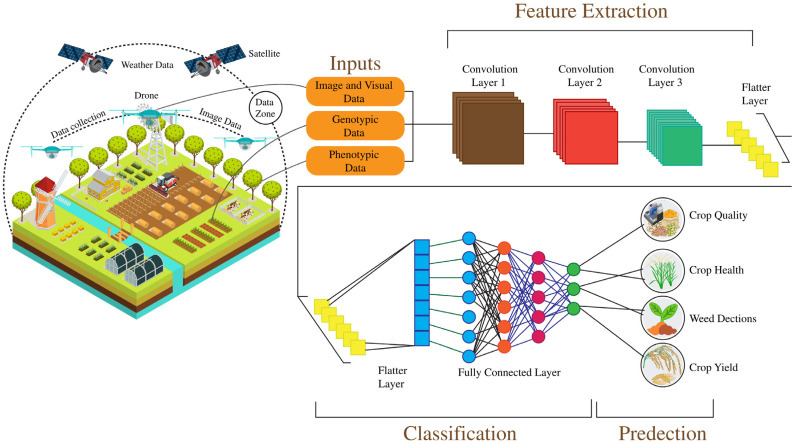
Demonstrates an intelligent agriculture framework integrating image, genotypic, and phenotypic data collected via drones, satellites, and weather systems. Using convolutional neural networks (CNNs), features are extracted through multiple layers for crop quality, crop health, weed detection, and crop yield prediction. The system highlights the role of deep learning in classification and prediction to optimize agricultural practices.

**Figure 7 plants-14-00671-f007:**
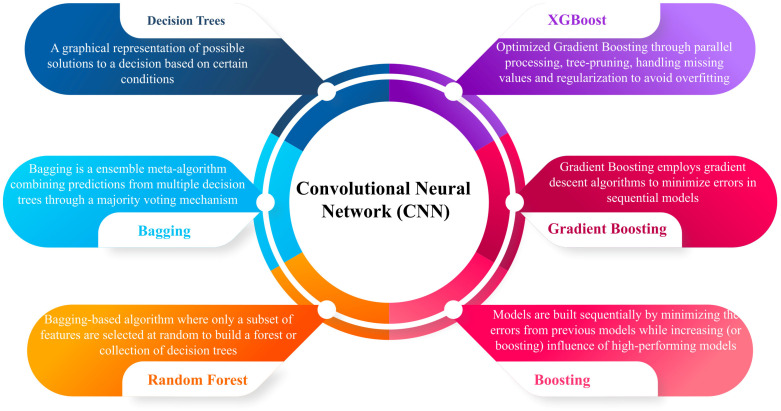
Presents key machine learning techniques used with Convolutional Neural Networks (CNNs), including Decision Trees, Bagging, Random Forest, Boosting, Gradient Boosting, and XGBoost. These methods optimize predictions by leveraging ensemble algorithms, gradient descent, and regularization to improve accuracy while minimizing errors and overfitting.

**Figure 8 plants-14-00671-f008:**
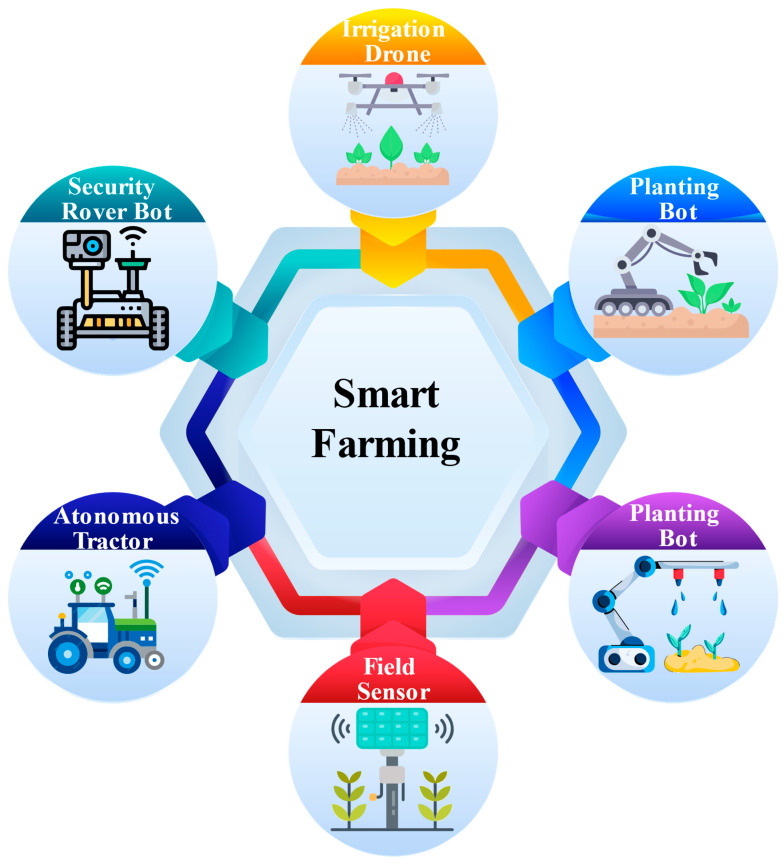
Showcases smart farming technologies, including irrigation drones, planting bots, autonomous tractors, security rover bots, and field sensors. These advanced tools optimize agricultural processes like planting, irrigation, monitoring, and security, enhancing efficiency and sustainability in modern farming.

**Figure 9 plants-14-00671-f009:**
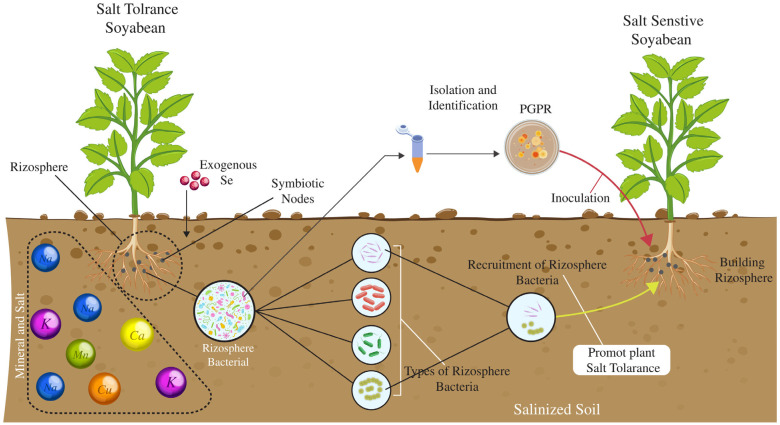
Illustrates the role of rhizosphere bacteria and PGPR (plant growth-promoting rhizobacteria) in enhancing salt tolerance in soybean plants. It highlights the recruitment of beneficial bacteria, exogenous selenium (Se) application, and symbiotic nodes to promote salt tolerance, contrasting salt-tolerant and salt-sensitive soybeans in salinized soil conditions.

**Table 1 plants-14-00671-t001:** Key genes and loci involved in plant stress tolerance.

Gene Name	Effect on Plant	Techniques	Tolerant Stress Type	Tissues	Ref.
*GmCLC1*	Enhances salinity tolerance by functioning as a chloride/proton antiporter	Expression analysis, Genetic mapping	Salinity	Roots, Leaves	[28]
*GmNHX1, GmNHX2*	Na^+^/H^+^ antiporters contributing to salt tolerance	Expression analysis, Genetic mapping	Salinity	Roots, All Organs (GmNHX2)	[29]
*GmsSOS1*	Na^+^ extrusion from roots regulates long-distance Na^+^ transport	Overexpression in Arabidopsis	Salinity	Roots, Shoots	[30]
*GmDREB2*	Binds to DRE motifs to enhance salinity tolerance	Transgenic plant generation	Salinity, Drought	Various Tissues	[31]
*GmERF*	Regulates downstream stress-responsive genes, enhances salt tolerance	Transgenic plant generation	Salinity, Drought	Various Tissues	[32]
*GmbZIP44, 62, 78, 110*	Improves salt and drought tolerance by regulating proline, Na^+^, and K^+^ levels	Expression analysis, Transgenic plants	Salt, Drought	Various Tissues	[32]
*GmWRKY12*	Regulates stress-responsive genes, involves salt and drought tolerance	Transcriptional profiling	Salt, Drought	Various Tissues	[32]
*GmMYB48*, *GmWD40*, *GmDHN15*, *GmGST1, GmLEA*	Upregulated in transgenic lines to enhance drought and salt tolerance	Transgenic plant generation, Expression analysis	Salt, Drought	Various Tissues	[33]
*Satt001*	Linked to drought tolerance via water-use efficiency	Genetic mapping, SSR markers	Drought	Various Tissues	[36]
*Satt002*	Associated with salinity tolerance via sodium ion exclusion	Genetic mapping, SSR markers	Salinity	Various Tissues	[36]
*Satt211*	Contributes to drought resistance via root system architecture	Genetic mapping, SSR markers	Drought	Roots	[37]
*Satt244*	Involved in osmotic adjustment under water-deficit conditions	Genetic mapping, SSR markers	Drought	Roots	[37]
*Satt312*	Linked to salt stress tolerance via ion transport regulation	Genetic mapping, SSR markers	Salinity	Various Tissues	[37]
*Satt337*	Associated with antioxidant enzyme activity under oxidative stress	Genetic mapping, SSR markers	Oxidative Stress (Salt)	Various Tissues	[37]
*GmCHX1*	Candidate gene for salt tolerance	Genome-wide sequencing, Fine mapping	Salinity	Roots	[40]
*GmSALT3*	Casual gene for salt tolerance on chromosome 3	Fine mapping, Genetic mapping	Salinity	Roots	[49]
*GmNCL*	Improves yield in salt-affected fields	Map-based cloning, Genetic mapping	Salinity	Various Tissues	[48]
*HaHB4*	Enhances drought tolerance, increases yield under water deficit	Transgenic plant generation, Molecular analysis	Drought	Leaves, Roots	[50]

## Data Availability

All the data is presented in this manuscript. For further details, consult the corresponding authors.

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
