# Peer review of "Integrative Approaches to Soybean Resilience, Productivity, and Utility: A Review of Genomics, Computational Modeling, and Economic Viability"

_plants, 2025, doi:10.3390/plants14050671_

Round 1

Reviewer 1 Report

Comments and Suggestions for Authors

1. Lack of detailed calculations regarding the economic costs and benefits of adopting the described technologies. Adding a more detailed economic study that compares the costs of implementing CRISPR and GWAS with the economic benefits achieved.

2. The article mentions sustainability, but does not go into sufficient detail about the ecological impact of soybean production for biofuels.  Including a comparative analysis of the ecological impact between various cultivation methods.

3. The possibility of risks associated with CRISPR or other advanced technologies is not addressed. Adding a section on the ethical challenges and potential risks associated with implementing these technologies.

4. references to be written in the same way appear in different writings.

Reviewer 2 Report

Comments and Suggestions for Authors

Dear Authors, 

The article is of interest, with a substantial number of publications (all literature items are referenced in the text). However, it is advisable to remove or summarise some extended paragraphs. Information that is considered to be common knowledge can also be removed from the manuscript. Some information is repetitive, and it may be worth combining or rearranging some content. The notes are attached in a PDF version.
